# STRUCTURED-INITIALIZATION LEARNING

## ABSTRACT

The emergence of large language models (LLMs) has revolutionized natural language processing, but their development and deployment face significant challenges in computational resources and environmental sustainability. Traditional self-supervised learning (SSL) paradigms requiring extensive computational infrastructure and exhibiting slow convergence rates, leading to increased energy consumption and longer training durations. While existing model fine-tuning techniques such as Low-Rank Adaptation (LoRA) are resource-intensive and fail to facilitate swift knowledge updates when integrating a mount of new data in model version iteration. To mitigate these challenges, we introduce **S**tructured-initi**ali**zation **l**earning (SAIL), a novel method for accelerating the training of neural network models by leveraging knowledge from (publicly available) pre-trained models. Our approach comprises two key components: (1) a parameter transformation technique that adjusts the dimensions of pre-trained model parameters to match the target architecture, and (2) a proximal parameter integration and retraining strategy that efficiently combines transformed parameters to initialize new models. We formalize the concept of Proximal Parameter and provide theoretical guarantees for its convergence advantages. Our approach achieves substantial reductions in training time and computational resources while maintaining or improving model performance on downstream tasks. These results indicate that SAIL provides a promising direction for the more efficient and accessible development of the deep learning community. Our code will be made publicly available.

## 1 INTRODUCTION

The emergence of Large Language Models (LLMs) such as GPT-3 (Brown et al., 2020), GPT-4 (OpenAI et al., 2023), PaLM (Chowdhery et al., 2023), and Gemini (Team et al., 2023) has ushered in a new era of natural language processing. These models have demonstrated unprecedented capabilities across a wide range of sequence tasks, showcasing remarkable advancements in representation learning. However, their success is accompanied by significant challenges. The development and deployment of LLMs require enormous computational resources, raising serious concerns about environmental sustainability and accessibility (Strubell et al., 2020). Moreover, while recent advancements have introduced efficient methods for data augmentation (Zhou et al., 2024) and synthesis of higher-quality data (Kaddour et al., 2023) to streamline the dataset, the pre-training process for these models remains prohibitively expensive and time-consuming (Sun et al., 2017).

A similar challenge constrains the advancement of self-supervised learning in both language and vision domains. Specifically, self-supervised learning for large models effectively leverages unlabeled data for representation learning (Liu, 2019; He et al., 2020; Chen et al., 2020), but it faces significant challenges in convergence speed and efficiency (Liu & Zhao, 2021; Wang et al., 2021). This paradigm renders the learning and fine-tuning process particularly resource-intensive (Faiz et al., 2023; Gao et al., 2020). As diverse new training data are curated, including high-quality synthetic data (Fan et al., 2024), the computational demands continue to escalate.

While techniques like LoRA and QLoRA (Hu et al., 2021; Dettmers et al., 2024; SONG et al., 2024) enable efficient fine-tuning of pre-trained models on domain-specific data, and model editing techniques (Meng et al., 2022) allow for rapid knowledge modification, these methods still demand substantial resources. Multiple modifications and edits can lead to model collapse (Wu & Papyan, 2024; Gu et al., 2024). Moreover, as indicated by Zhu & Li (2023), post-training fine-tuning struggles to rectify erroneous knowledge learned during the training phase, potentially perpetuating hallucina-

Figure 1: **The Structured Initialization Learning framework.** Our method leverages diverse pre-trained models from open-source platforms, applying weight linear transformations to adapt them to the target model size. We then merge these transformed models through parameter aggregation, creating a informative initialization ($\theta^P$). This amalgamated starting point serves as the initial parameters in the Loss Space, enabling a more efficient optimization trajectory towards the optimal parameters ($\theta^\star$) for the new target model. This unified framework can facilitate rapid model iteration on new datasets while harnessing the advantages of pre-trained models, leading to faster convergence to the $\theta^\star$.

tions (Ji et al., 2023) and persistent model biases (Blodgett et al., 2020). Furthermore, fine-tuning is typically infeasible when dealing with changes to model architecture (Dettmers et al., 2024). Consequently, the current strategy for updating model versions with architectural, capacity, and knowledge modifications——such as progressing from Llama 1 to Llama 2/3 (Touvron et al., 2023)——involves training new models from scratch using large volumes of fresh data through repeated training cycles.

Building upon the preceding analysis, we propose that a key computational challenge in current model training methods lies in their failure to effectively leverage knowledge from cross-architectural and cross-domain pre-trained models, instead relying solely on training from randomly initialized models. The question then arises: *how can we effectively utilize pre-trained models to initially tap into their existing knowledge, followed by efficient new training or continued training?*

In this paper, we aim to harness pre-trained models to obtain an initialized parameter that is proximal to optimal parameter of model that accelerates the training of a new large model, facilitating easier adaptation to new data and techniques. To achieve this, we first need to transform the parameters of previously trained models to match the parameter size and architecture of our new target model. We then need to find an optimal integration of these parameters to form the proximal parameter.

To address these challenges and bridge the gap, we introduce an innovative training paradigm:

> **Proposition 1 (Accelerating task-agnostic training via pre-trained model knowledge) .** *Let $M = \{\psi_1, \ldots, \psi_k, \psi_K\}$ be a set of models pre-trained over datasets $S = \{D_1, \ldots, D_K\}$, $D$ a new dataset, $\phi_{\theta^{(0)}}$ a initialized model, and $\mathscr{L}_D$ a training process with $T$ steps on data $D$. We propose a parameter initializer $\mathscr{P}$ centered on $\theta$, aimed at improving training efficiency such that:*
>
> $$\mathcal{L}_D(\theta^{(T)} \leftarrow \mathscr{L}_D(\theta^{(0)} \leftarrow \mathscr{P}(M, S))) < \mathcal{L}_D(\theta^{(T)} \leftarrow \mathscr{L}_D(\theta^{(0)})), \tag{1}$$
>
> *where $\mathcal{L}_D$ represents the loss function evaluating performance on data $D$.*

To investigate our new learning paradigm claim in Proposition 1 , we introduce SAIL, a novel method that leverages freely available pre-trained models to accelerate training. Our approach improves efficiency in the initial training stages by effectively utilizing the parameters of pre-trained models directly, thus establishing a rapid pathway for representation model training (see Figure 1 ).

The core of our method involves inheriting and integrating knowledge directly from pre-trained model parameters, creating a shortcut in the learning process. This approach allows the initial model $\phi$ to effectively reach a "Proximal Parameter" $\theta^P$ that is closer to optimal than randomly initialized parameters, thereby significantly accelerating the learning process. As formalized in Definition 1 , our method first aligns the parameter dimensions from various pre-trained models into a unified format. Subsequently, we execute a weighted parameter averaging that accounts for the effective knowledge embedded in the parameters of each model, thereby enhancing both knowledge transfer and representation learning efficiency.

This framework leverages the extensive range of publicly available pre-trained models, providing a novel paradigm for representation learning and notably expediting the model development process. Our main contributions are as follows:

(a) We introduce SAIL, a novel method for accelerating the training of large language models by leveraging knowledge from pre-trained models. This approach includes a parameter transformation technique and a proximal parameter integration strategy, effectively utilizing the wealth of publicly available models (see Section 4).

(b) We provide theoretical foundations for our method, including the formalization of the Proximal Parameter concept and convergence guarantees. Our analysis demonstrates how SAIL leads to faster convergence compared to random initialization (see Section 3 and Appendix A).

(c) We conduct extensive experiments across multiple modalities, including natural language processing and computer vision tasks and various model architectures. Our results show that SAIL not only accelerates training on its own but also demonstrates consistent performance improvements across different datasets, model sizes, and learning paradigms (supervised and self-supervised). This versatility is evidenced by experiments on GPT-2 variants for NLP and ResNet architectures for image classification, showcasing the broad applicability of our method. (see Section 4.4 and Section 4.5).

## 2 RELATED WORK

Our work on SAIL builds upon and extends several areas of research in efficient training techniques, particularly for LLMs. We discuss three distinct relevant areas: 1) techniques for efficient training; 2) methods for transforming and reusing deep models; and 3) model merging methods to combine different models.

**Efficient training techniques for representation learning.** A critical aspect of efficient training involves effective model initialization, which can significantly influence convergence speed and overall training efficiency. Techniques such as Xavier Initialization (Glorot & Bengio, 2010) and Kaiming Initialization (He et al., 2015) have been foundational in ensuring stable gradients and accelerating convergence during training.

Beyond initialization, dynamic architecture approaches achieve efficiency by dynamically activating or deactivating network components during training, employing strategies such as layer stacking (Gong et al., 2019), layer dropping (Zhang & He, 2020), and the use of sparse attention mechanisms (Child et al., 2019). Batch selection techniques enhance learning efficiency by prioritizing the most informative training examples, utilizing methods like selective backprop (Jiang et al., 2019), RHO loss (Mindermann et al., 2022), and curriculum learning (Bengio et al., 2009). Furthermore, innovative optimizers such as Lion (Wang et al., 2023a), Sophia (Liu et al., 2023), and AdaFactor (Shazeer & Stern, 2018) provide alternatives to traditional optimizers like Adam(W), promoting more efficient convergence. Techniques like mixed-precision training (Micikevicius et al., 2017) and gradient checkpointing (Chen et al., 2016) further mitigate computational demands by reducing memory consumption, thereby enabling the training of larger models on limited hardware resources.

Unlike these methods, our SAIL directly leverages the parameters of multiple pre-trained models to create a well-informed starting point, potentially reducing the need for complex training optimizations. While these existing techniques could be combined with our approach for further efficiency gains.

**Model reuse and expansion.** Approaches in this category focus on leveraging pre-existing knowledge to initialize or expand models. Model reuse methods enable the adaptation of pre-trained models for new tasks or larger architectures without retraining from scratch. Notable examples include Samragh et al. (2024), who explore scalable model reuse strategies, and Wang et al. (2023b), who investigate data-driven approaches for model adaptation.

Model expansion techniques aim to scale smaller models to initialize larger ones, ensuring that the expanded models retain the learned representations. Classic methods like Net2Net (Chen et al., 2015) provide a foundation for expanding neural networks by transferring knowledge from smaller to larger architectures. More recent advancements, such as Learning to Grow (Wang et al., 2023a) and MorphNet (Gordon et al., 2018), focus on dynamically increasing model capacity during training, thereby enhancing scalability and performance.

Progressive learning methods gradually increase model capacity during training, which can lead to more efficient learning and better generalization. Works by Li et al. (2022); Pan et al. (2024) introduce automated strategies for progressive model scaling.

Knowledge transfer techniques utilize distillation to transfer knowledge from smaller to larger models or between models of similar sizes, enhancing performance and training efficiency. Methods such as Knowledge Inheritance (Qin et al., 2021) and Born-Again Networks (Furlanello et al., 2018) exemplify effective strategies for transferring learned representations.

Our research focuses on the novel capability to incorporate parameters from multiple pre-trained models with diverse architectures. This approach generates a sophisticated initialization for the target model, surpassing conventional methods that are limited to single-model adaptation or expansion.

**Model merging.** This area focuses on combining multiple models to create a single, more powerful model. Simple approaches like Model Soup (Wortsman et al., 2022) apply straightforward weight averaging to merge models, thus combining their diverse learned representations. Advancements such as Checkpoint Merging (Liu et al., 2024) introduce Bayesian optimization to effectively select and weight various checkpoints, resulting in a more robust and high-performing merged model. Additionally, techniques like cross-model integration via MindMerger (Huang et al., 2024) enable the fusion of models with varying specializations, enhancing the overall capabilities of the merged system. Dynamic expert merging methods, including DELLA-Merging (Tej Deep et al., 2024), integrate specialized expert models dynamically, allowing the merged model to adapt to a variety of tasks during inference. Adaptive weighting approaches such as AdaMerging (Yang et al., 2023) and MetaGPT (Zhou et al., 2024) leverage dynamic weighting schemes and meta-learning to fine-tune the merging process, ensuring optimal integration of constituent models' strengths. Furthermore, task-oriented merging strategies like Task Arithmetic (Ilharco et al., 2022), Language and Task Arithmetic (Chronopoulou et al., 2023), and Task Arithmetic in Tangent Space (Ortiz-Jimenez et al., 2024) focus on blending models trained on different tasks, thereby creating versatile LLMs adept at multiple applications.

## 3 Accelerated Training via Proximal Parameter

In this section, we formalize the problem of accelerating the training of large auto-regressive language models (LLMs) by leveraging knowledge from pre-trained models. We introduce the concept of *Proximal Parameter*, which serves as the foundation for our acceleration technique. We present rigorous mathematical definitions and theorems illustrating the accelerated convergence benefits of using proximal parameter initialization for model training.

### 3.1 Proximal Parameter

Let $\phi_{\boldsymbol{\theta}} : \mathcal{X} \to \mathcal{Y}$ denote an model parameterized by $\boldsymbol{\theta} \in \mathbb{R}^d$, where $\mathcal{X}$ is the input space and $\mathcal{Y}$ is the output space. Let $\ell : \mathcal{Y} \times \mathcal{Y} \to \mathbb{R}_{\geq 0}$ be a loss function measuring the discrepancy between the output of model and the target output. Our goal is to minimize the expected loss $\mathbb{E}\left[\ell(\phi_{\boldsymbol{\theta}}(\mathbf{x}), \mathbf{y})\right]$:

$$\boldsymbol{\theta}^{\star} = \arg\min{}_{\boldsymbol{\theta}}\{\mathcal{J}_D(\boldsymbol{\theta})\} = \arg\min{}_{\boldsymbol{\theta}}\{\mathbb{E}_{(\mathbf{x},\mathbf{y})\sim D}\left[\ell(\phi_{\boldsymbol{\theta}}(\mathbf{x}), \mathbf{y})\right]\}, \quad (2)$$

where $(\mathbf{x}, \mathbf{y})$ are input-output pairs sampled from real data distribution $D$ and $\mathbf{y} \in \mathcal{Y}$.

To accelerate convergence during training, we aim to find an effective initialization for the model parameters $\boldsymbol{\theta}$. The key insight is that we can leverage the knowledge encoded in multiple pre-trained models to construct a more informed starting point for training the new model. We now introduce the concept of *Proximal Parameter*, which represents an aggregation of knowledge from multiple pre-trained models, adjusted to match the architecture and knowledge of the target model $\phi_{\boldsymbol{\theta}^{\star}}$.

> **Definition 1 (Proximal Parameter).** *Let $\{\boldsymbol{\theta}_1, \boldsymbol{\theta}_2, \ldots, \boldsymbol{\theta}_K\}$ be a set of $K$ parameter vectors from pre-trained models $M$, where each $\boldsymbol{\theta}_i \in \mathbb{R}^{d_i}$. Define $\boldsymbol{T}_i : \mathbb{R}^{d_i} \to \mathbb{R}^d$ as a transformation function mapping each $\boldsymbol{\theta}_i$ to the parameter space $\mathbb{R}^d$ of the target model $\phi_{\boldsymbol{\theta}^{\star}}$. The proximal parameter $\boldsymbol{\theta}^{\mathrm{P}} \in \mathbb{R}^d$ is the optimal linear combination of the transformed parameters, weighted by $\gamma_i$, defined*

as $\boldsymbol{\theta}^{\mathrm{P}} = \sum_{i=1}^{K} \gamma_i^\star \tilde{\boldsymbol{\theta}}_i$ based on the loss function $\mathcal{J}_D$ and training process $\mathscr{L}_D$ such that:

$$\gamma_1^\star, \ldots, \gamma_K^\star = \arg\min_{\gamma_1, \ldots, \gamma_K} \{ \mathcal{J}_D(\mathscr{L}_D(\sum_{i=1}^{K} \gamma_i \tilde{\boldsymbol{\theta}}_i)) \}. \tag{3}$$

However, calculating (3) poses a challenge due to the nonlinear properties of $\mathcal{J}_D$ and $\mathscr{L}_D$. Alternatively, define the proximal parameter based on Frobenius norm in parameter space as:

$$\gamma_1^\star, \ldots, \gamma_K^\star = \arg\min_{\gamma_1, \ldots, \gamma_K} \left\| \sum_{i=1}^{K} \gamma_i \tilde{\boldsymbol{\theta}}_i - \boldsymbol{\theta}^\star \right\|_F^2, \tag{4}$$

where the transformed parameters are defined as $\tilde{\boldsymbol{\theta}}_i = \boldsymbol{T}_i(\boldsymbol{\theta}_i) \in \mathbb{R}^d$ for $i = 1, \ldots, K$.

The proximal parameter $\boldsymbol{\theta}^{\mathrm{P}}$ aggregates information from multiple pre-trained models into a single set of parameters, serving as an informed initialization for the target model.

## 3.2 CONVERGENCE ANALYSIS

Before presenting the theorem, we introduce key assumptions that underpin our analysis. We concentrate on linear models, assuming that all pre-trained models share an identical architecture. Under these conditions, any variation among the models stems solely from differences in their training datasets. Additionally, in linear models, the parameters are uniquely determined by the training data. These assumptions allow us to quantify model proximity by examining dataset differences, offering a coherent framework for comparing pre-trained models with randomly initialized ones.

**Theorem 1 (Proximity-based model initialization advantage,** *proof in* **Appendix A ).** *For any proportionality factor $\alpha \in (0,1)$, the squared Euclidean distance between the pre-trained model parameters $\boldsymbol{\theta}_i$ and the target parameters $\boldsymbol{\theta}^\star$ satisfies the following probabilistic bound:*

$$\Pr\left( \|\boldsymbol{\theta}_i - \boldsymbol{\theta}^\star\|_2^2 \leq \alpha \|\boldsymbol{\theta}_{rand} - \boldsymbol{\theta}^\star\|_2^2 \right) \geq 1 - O\left( \frac{\tau^2 + \beta}{\alpha} \right), \tag{5}$$

*Here, $\boldsymbol{\theta}_{rand}$ represents the randomly initialized model parameters, $\tau$ quantifies the variance of the pre-training dataset mean difference compared to the target dataset, while $\beta$ represents the upper bound on the variance of the perturbation in the pre-training dataset's variance. Smaller values of $\tau$ and $\beta$ reflect greater proximity between $\boldsymbol{\theta}_i$ and $\boldsymbol{\theta}^\star$.*

Theorem 1 shows that, with high probability, pre-trained parameters $\boldsymbol{\theta}_i$ are closer to the optimal target parameters $\boldsymbol{\theta}^\star$ than randomly initialized parameters, especially when the pre-training dataset distribution $D_i$ is statistically similar to the target $D^\star$.

**Theorem 2 (Convergence of proximal parameter initialization,** *proof in* **Appendix B ).** *Let $\{\boldsymbol{\theta}^{(t)}\}$ be the sequence of parameters generated by gradient descent with fixed learning rate $\eta \in \left(0, \frac{1}{L}\right)$, initialized at $\boldsymbol{\theta}^{(0)} = \boldsymbol{\theta}^P = \sum_{i=1}^{n} \gamma_i^\star \tilde{\boldsymbol{\theta}}_i$, where $\boldsymbol{\theta}^P$ is defined as in (3). Then, the suboptimality at iteration $T$ satisfies:*

$$\mathcal{J}_D(\boldsymbol{\theta}^{(T)}) - \mathcal{J}_D(\boldsymbol{\theta}^\star) \leq (1 - \eta\mu)^T \left( \mathcal{J}_D(\boldsymbol{\theta}^P) - \mathcal{J}_D(\boldsymbol{\theta}^\star) \right), \tag{6}$$

*where $L > 0$ is the Lipschitz constant of the gradient of the loss function $\mathcal{J}_D(\boldsymbol{\theta})$, and $\mu > 0$ is the strong convexity parameter of $\mathcal{J}_D(\boldsymbol{\theta})$. Furthermore, we have:*

$$\mathcal{J}_D(\boldsymbol{\theta}^P) - \mathcal{J}_D(\boldsymbol{\theta}^\star) \leq \frac{L}{2} \left\| \boldsymbol{\theta}^P - \boldsymbol{\theta}^\star \right\|_2^2. \tag{7}$$

*By choosing the weights $\gamma_i^\star$ to minimize $\left\| \boldsymbol{\theta}^P - \boldsymbol{\theta}^\star \right\|_2$, we effectively minimize the bound on the initial suboptimality, leading to faster convergence compared to random initialization.*

In light of Theorem 2, we propose that initializing with the proximal parameter $\boldsymbol{\theta}^P$ is likely to lead to faster convergence compared to random initialization. Specifically, Theorem 2 shows that the convergence rate of the loss function can be controlled by the initial parameter distance, while Theorem 1 demonstrates that the distance between the proximal parameter $\boldsymbol{\theta}^P$ and the

optimal parameter $\boldsymbol{\theta}^\star$ is, with high probability, smaller than that of randomly initialized parameters. Therefore, by combining these results, we can assert that, with high probability, initialization with the proximal parameter $\boldsymbol{\theta}^P$ leads to faster convergence compared to random initialization. A detailed proof of this argument can be found in Appendix C. Moreover, by weighting the contributions of each transformed parameter, we can prioritize models closer to the target. This strategy ensures that the optimization process starts from a point nearer to the global optimum, thereby enhancing the overall convergence rate of the gradient descent algorithm.

## 4 STRUCTURED-INITIALIZATION LEARNING

In this section, we introduce SAIL, a novel approach that accelerates the training of large language models by directly leveraging the parameters of pre-trained models. Traditional methods, such as knowledge distillation, focus on aligning model outputs or hidden states, often neglecting the rich information embedded in the model parameters themselves. We posit that the parameters of a model encapsulate compressed knowledge acquired during training, and different models may provide diverse perspectives even when trained on similar data. By directly utilizing these parameters, we aim to create an effective starting point for training new models, leading to faster convergence and improved performance.

Our methodology comprises two main components: (1) *Parameter Transformation*, where we adjust the dimensions of pre-trained model parameters to match the target model architecture, and (2) *Proximal Parameter Integration and Retraining*, where we integrate the transformed parameters to initialize the new model and continue training on new data.

### 4.1 PARAMETER TRANSFORMATION

To harness the knowledge embedded in pre-trained models, we first transform their parameters to be compatible with the target model's architecture. This involves adjusting both the *width* (dimensionality of layers) and the *depth* (number of layers) of the models.

**Width transformation.** For each layer in the model, we define a *width transformation* function that maps the parameters from the source dimensionality to the target dimensionality. Given a weight matrix $\boldsymbol{\theta} \in \mathbb{R}^{d_{\text{in}} \times d_{\text{out}}}$ from a pre-trained model, we aim to transform it into a matrix $\tilde{\boldsymbol{\theta}} \in \mathbb{R}^{d'_{\text{in}} \times d'_{\text{out}}}$ that aligns with the target model's dimensions.

$$\tilde{\boldsymbol{\theta}} = \begin{bmatrix} \mathbf{c}_{11} & \mathbf{c}_{12} & \cdots & \mathbf{c}_{1d_{\text{in}}} \\ \mathbf{c}_{21} & \mathbf{c}_{22} & \cdots & \mathbf{c}_{2d_{\text{in}}} \\ \vdots & \vdots & \ddots & \vdots \\ \mathbf{c}_{d'_{\text{in}}1} & \mathbf{c}_{d'_{\text{in}}2} & \cdots & \mathbf{c}_{d'_{\text{in}}d_{\text{in}}} \end{bmatrix} \mathbf{D} \begin{bmatrix} \mathbf{c}'_{11} & \mathbf{c}'_{12} & \cdots & c'_{1d'_{\text{out}}} \\ \mathbf{c}'_{21} & \mathbf{c}'_{22} & \cdots & c'_{2d'_{\text{out}}} \\ \vdots & \vdots & \ddots & \vdots \\ \mathbf{c}'_{d_{\text{out}}1} & \mathbf{c}'_{d_{\text{out}}2} & \cdots & \mathbf{c}'_{d_{\text{out}}d'_{\text{out}}} \end{bmatrix}^\top \tag{8}$$

where $\mathbf{c}_{\text{in}} \in \mathbb{R}^{d'_{\text{in}} \times d_{\text{in}}}$ and $\mathbf{c}_{\text{out}} \in \mathbb{R}^{d'_{\text{out}} \times d_{\text{out}}}$ are transformation matrices that map dimensions from the source to the target. This mapping can be learned or defined using schemes such as random projection or interpolation, followed by normalization to ensure numerical stability.

**Depth transformation.** To adjust the number of layers, we introduce a *depth transformation* function that combines or splits the parameters of layers. Given $L$ layers in the pre-trained model and $L'$ layers in the target model, we define:

$$\tilde{\boldsymbol{\theta}}^k = \begin{bmatrix} d_{k1} & d_{k2} & \cdots & d_{kL} \end{bmatrix} \begin{bmatrix} \boldsymbol{\theta}^1 \\ \boldsymbol{\theta}^2 \\ \vdots \\ \boldsymbol{\theta}^L \end{bmatrix}, \quad \text{for } k = 1, \ldots, L' \tag{9}$$

Here, $\tilde{\boldsymbol{\theta}}^k$ represents the parameters of the $k$-th layer in the target model. The transformation is defined as a linear combination of the source model's layer parameters $\boldsymbol{\theta}^i$ ($i = 1, \ldots, L$). The coefficient matrix $\mathbf{D}_{\text{depth}} = [d_{ki}] \in \mathbb{R}^{L' \times L}$ controls this linear combination. For each row $k$ of $\mathbf{D}_{\text{depth}}$ corresponds to a layer in the target model. Each column $i$ corresponds to a layer in the source model. The element $d_{ki}$ represents the contribution of the $i$-th source layer to the $k$-th target layer.

## 4.2 PROXIMAL PARAMETER INTEGRATION AND RETRAINING

After transforming the parameters of the pre-trained models to match the target architecture, we integrate them to form the proximal parameter $\boldsymbol{\theta}^{\mathrm{P}}$ as defined in (3).

**Integration of transformed parameters based on total variation distance.** Using the transformed parameter $\tilde{\boldsymbol{\theta}}_i$, we compute the proximal parameter by *approximately optimal* weights $\gamma_i^\star$, i.e., $\boldsymbol{\theta}^{\mathrm{P}} = \sum_{i=1}^n \gamma_i^\star \tilde{\boldsymbol{\theta}}_i$. Specifically, let $\mathrm{D}_{\mathrm{TV}}(P, Q)$ denote the total variation distance between two distributions. Building on Theorem 3, we can compute the approximately optimal weights $\gamma_i^\star$.

---

**Theorem 3 (Optimal combination coefficients, *proof in Appendix E*).** *Given the proximal parameter $\boldsymbol{\theta}^{\mathrm{P}} \in \mathbb{R}^d$, defined as the weighted convex combination of transformed parameters:*

$$\boldsymbol{\theta}^{\mathrm{P}} = \sum_{i=1}^n \gamma_i^\star \tilde{\boldsymbol{\theta}}_i, \quad \text{where} \quad \sum_{i=1}^n \gamma_i^\star = 1, \quad \gamma_i^\star \geq 0, \tag{10}$$

*the optimal combination coefficients $\boldsymbol{\gamma}^\star = [\gamma_1^\star, \gamma_2^\star, \ldots, \gamma_n^*]^\top$ that minimize the distance between $\boldsymbol{\theta}^{\mathrm{P}}$ and the target parameter $\boldsymbol{\theta}^\star$ are as follows:*

*(a) **Case** $n = 2$: When there are two pre-trained models, the optimal combination coefficients $\gamma_1^\star$ and $\gamma_2^\star$ can be explicitly determined under the constraint $\gamma_i^\star \geq 0$:*

$$\gamma_1^\star = \frac{\mathrm{D}_{TV}(D_2, D^\star)^2 + \mathrm{D}_{TV}(D_1, D_2)^2 - \mathrm{D}_{TV}(D_1, D^\star)^2}{2\mathrm{D}_{TV}(D_1, D_2)^2}, \quad \gamma_2^\star = 1 - \gamma_1^\star, \tag{11}$$

*This solution is optimal provided that $\gamma_1^\star, \gamma_2^\star \geq 0$.*

*(b) **Case** $n > 2$: For more than two pre-trained models, an explicit solution for the optimal combination coefficients $\boldsymbol{\gamma}^\star$ generally does not exist under the constraints $\gamma_i^\star \geq 0$. However, if we further impose $\gamma_i^\star > 0$ for all $i$, the optimal coefficients can be explicitly determined as:*

$$\boldsymbol{\gamma}^\star = \frac{\mathbf{H}^{-1}\mathbf{e}}{\mathbf{e}^\top \mathbf{H}^{-1}\mathbf{e}}, \tag{12}$$

*where $\mathbf{H} \in \mathbb{R}^{n \times n}$ is a matrix with elements $H_{ij}$ determined by:*

$$H_{ij} = \mathrm{D}_{TV}(D_i, D^\star)^2 + \mathrm{D}_{TV}(D_j, D^\star)^2 - \mathrm{D}_{TV}(D_i, D_j)^2. \tag{13}$$

*and $\mathbf{e}$ is an $n$-dimensional vector with all entries equal to 1.*

---

Furthermore, Theorem 3 reveals an important insight: the smaller the total variation distance $\mathrm{D}_{\mathrm{TV}}(D_i, D^\star)$, the larger the corresponding weight $\gamma_i^\star$. In other words, pre-trained models closer to the target distribution receive higher weights in the optimal combination, ensuring that these models contribute more significantly to the proximal parameter and improve the approximation accuracy.

**Retraining on new data.** With the proximal parameter $\boldsymbol{\theta}^{\mathrm{P}}$ as the initial parameter of model $\phi$, we proceed to retrain the model $\phi$ on new data $D$. The training objective is to minimize the expected loss $\mathcal{J}(\boldsymbol{\theta})$ as in (2). Starting from $\boldsymbol{\theta}^{(0)} = \boldsymbol{\theta}^{\mathrm{P}}$, the model is expected to converge faster due to the informative initialization, as demonstrated in Theorem 2. Furthermore, its generalization error remains bounded as shown in Theorem 4.

## 4.3 EXPERIMENTAL SETTING

In this section, we provide a comprehensive overview of our experimental setup, encompassing the models, datasets, baselines, metrics, and training details used to evaluate the effectiveness of our proposed SAIL method across different modalities, including natural language processing and computer vision tasks. Additional implementation details, including specific hyperparameters, and detailed model architectures, are provided in Appendix H.

**Base Models.** For the natural language processing sequence modeling task, we utilize the GPT-2 architecture (Radford et al., 2019), employing the nanoGPT[1] implementation. We consider models of

---

[1]https://github.com/karpathy/nanoGPT

varying sizes to assess the scalability of our method. Specifically, our base experiment configuration includes models with 6 layers and a hidden dimension of $384$, amounting to approximately $21$ million parameters. For the computer vision task, we employ convolutional neural network architectures, focusing on ResNet variants (He et al., 2016). We use the standard ResNet-18 models to evaluate the applicability of our method in the vision domain. Additionally, we consider modified version of ResNet-18 and ResNet-34. See more detailed configurations in Appendix H

**Datasets.** In the NLP domain, we use the OPENWEBTEXT (Gokaslan et al., 2019) and WIKITEXT-103 (Merity et al., 2016) datasets for training and evaluation. These datasets are partitioned to simulate different training subsets, allowing us to train multiple models on different data partitions for the Proximal Parameter method. In the computer vision domain, we conduct experiments on standard image classification datasets: CIFAR-10, CIFAR-100 (Krizhevsky et al., 2009), and Tiny ImageNet (Le & Yang, 2015).

**Methods.** We pre-train ResNet models using both supervised and self-supervised learning paradigms, with the latter employing the BYOL framework (Grill et al., 2020).

**Metrics.** We measure performance using **top-1** and **top-5 accuracy** on the validation sets of CIFAR-10, CIFAR-100, and Tiny ImageNet. We also monitor training loss and convergence rates to assess the efficiency of different training methods.

### 4.4 EMPIRICAL EVALUATION OF SAIL'S EFFICACY

To evaluate the robustness and effectiveness of our SAIL method across different data distributions and scenarios, we conducted a series of experiments focusing on data distribution effects, overlap impacts, and cross-dataset generalization.

**Dataset Partitioning and Distributional Analysis** We partitioned our dataset into three distinct subsets ($D_1$, $D_2$, and $D_t$) based on feature segmentation using mean token values. This partitioning strategy, inspired by the theoretical foundations of our method, allows us to simulate diverse data distributions commonly encountered in real-world scenarios. We computed the mean token value for blocks of data and used the 33rd and 66th percentiles as thresholds, a choice motivated by our aim to create balanced yet distinct subsets. Samples with mean token values below the lower threshold were assigned to $D_1$, those between the thresholds to $D_2$, and those above the upper threshold to $D_t$, resulting in three datasets with distinct distributions that serve as an ideal testbed for our SAIL method.

As illustrated in Figure 2a, a t-SNE visualization shows that these datasets form three clearly separated clusters, empirically confirming the effectiveness of our theoretically-motivated feature-based splitting approach.

To demonstrate the superiority of SAIL over random initialization in practice, we trained two foundation models, $\boldsymbol{\theta}_1$ and $\boldsymbol{\theta}_2$, on $D_1$ and $D_2$, respectively. Our objective was to merge these models using our SAIL method and evaluate the convergence speed of the merged model on the new data partition $D_t$, thereby testing the practical implications of our theoretical framework. As shown in Figure 2b, the validation loss for random initialization is 10.8866, significantly higher than the minimum loss of 4.9782 achieved by our SAIL method. This substantial gap not only demonstrates the effectiveness of our approach in providing a more favorable initialization point in the loss landscape but also validates our theoretical predictions about the benefits of informed parameter initialization.

We systematically explored the parameter space by merging $\boldsymbol{\theta}_1$ and $\boldsymbol{\theta}_2$ using 30 values of $\gamma$ in the range $[-1, 2]$ with increments of $0.1$. This comprehensive sweep allows us to empirically validate the theoretical predictions of the optimal combination coefficient $\gamma^*$ as detailed in Theorem 3. Each merged model was then retrained on $D_t$ for a small number of iterations (50, 100, and 200), and we recorded the validation loss, providing insights into both short-term and longer-term effects of our initialization strategy.

Figure 2b presents the validation loss after 50, 100, and 200 iterations of retraining for different values of $\gamma$. The optimal $\gamma$ corresponds to the minimum validation loss, indicating the best merging ratio

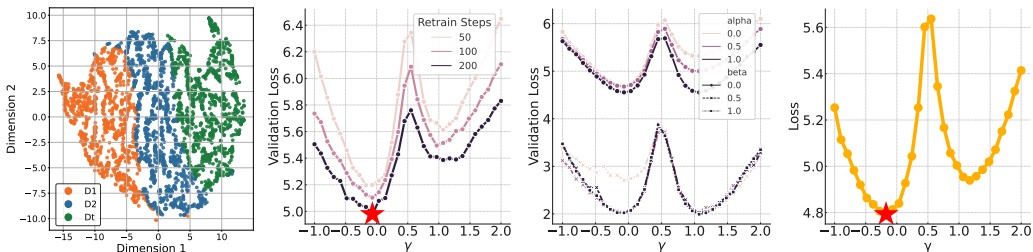

(a) t-SNE Visualization of $D_1$, $D_2$, and $D_t$    (b) Validation Loss vs. $\gamma$    (c) Validation Loss vs. $\gamma$ (Overlaps)    (d) Cross-Dataset Validation Loss

Figure 2: **Experimental Results:** (a) t-SNE visualization of datasets $D_1$, $D_2$, and $D_t$. (b) Validation loss after retraining as a function of merge ratio $\gamma$. (c) Validation loss on $D_t$ vs. $\gamma$ for various data overlaps. (d) Cross-dataset validation loss on WikiText-103.

for rapid convergence on $D_t$. This empirical finding aligns closely with our theoretical predictions, further validating the robustness of our approach.

Applying Theorem 3 to our experimental scenario, we compute the theoretical optimal $\gamma^*$ using the Maximum Mean Discrepancy (MMD) distances between datasets: $\gamma^* = \frac{\text{MMD}(D_1,D_t)^2 + \text{MMD}(D_1,D_2)^2 - \text{MMD}(D_2,D_t)^2}{2 \cdot \text{MMD}(D_1,D_2)^2} = -0.1244$. This theoretically derived $\gamma^*$ value of -0.1244 closely aligns with the empirically observed optimal $\gamma$ in Figure 2b, providing strong evidence for the practical applicability of our theoretical framework.

**Impact of Data Overlap on SAIL**    To investigate the impact of data overlap, we defined parameters $\alpha$ and $\beta$. Here, $\alpha$ represents the overlap between $D_1$ and $D_2$, while $\beta$ denotes the overlap of both $D_1$ and $D_2$ with $D_t$. These overlap fractions range from 0.0 to 1.0, encompassing scenarios from no overlap to full overlap.

For each combination of $\alpha$ and $\beta$, we trained separate models $\boldsymbol{\theta}_1$ and $\boldsymbol{\theta}_2$ on $D_1$ and $D_2$, respectively, and an optimal model $\boldsymbol{\theta}^\star$ on the target dataset $D_t$. We evaluated 30 equally spaced $\gamma$ values in the range $[-1, 2]$ for merging $\boldsymbol{\theta}_1$ and $\boldsymbol{\theta}_2$. The merged models were then fine-tuned on $D_t$ for 50 steps to assess the impact of additional training.

Our analysis revealed several key findings, as shown in Figure 2c:

(a) The optimal $\gamma$ values concentrating near 0.0 and 1.0 suggest that the most effective interpolation often involves one model being slightly regularized by another. It is important to note that when $\gamma$ is 0.5, the observed spike in performance does not necessarily indicate a rapid convergence to the global optimum.
(b) As the overlap between datasets increased, the validation loss curves exhibited greater symmetry around the optimal $\gamma$ value. This symmetry suggests that both component models extracted similar features from the overlapping data.
(c) At lower overlap fractions, the asymmetry observed in the curves implies that one of the component models captured more generalizable features relevant to the target dataset $D_t$.

These observations are consistent with our theoretical considerations in Assumption 4, where the relationship between model parameters and data features influences the optimal merging strategy.

**Cross-Dataset Transfer Analysis**    To assess SAIL's ability to leverage cross-dataset knowledge, we conducted experiments transferring models trained on OpenWebText to the WikiText-103 dataset. This setup tests the algorithm's capacity to generalize across datasets with different styles, vocabularies, and content structures.

We first trained two models, $\boldsymbol{\theta}_1$ and $\boldsymbol{\theta}_2$, on disjoint partitions $D_1$ and $D_2$ of OpenWebText. These partitions were created using a feature-based splitting approach that considers the mean token value of samples. We then used SAIL to initialize a model for fine-tuning on WikiText-103, which serves as our target dataset $D_t$.

Figure 2d presents the validation loss on WikiText-103 as a function of the merge ratio $\gamma$. The curve's shape and the existence of a clear optimal $\gamma$ demonstrate SAIL's capacity to effectively

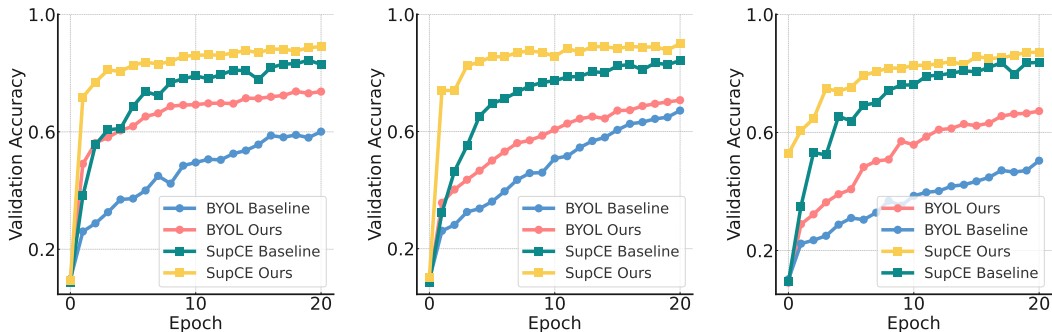

(a) ResNet-18 Modified (CIFAR-10)  (b) Standard ResNet-18 (CIFAR-10)  (c) ResNet-34 Modified (CIFAR-10)

Figure 3: **Accuracy in Different ResNet Configurations: (a)** Accuracy of ResNet-18 Modified trained with BYOL and SupCE on CIFAR-10. **(b)** Accuracy of standard ResNet-18 trained with BYOL and SupCE on CIFAR-10. **(c)** Accuracy of ResNet-34 Modified trained with BYOL and SupCE on CIFAR-10.

combine knowledge from disparate datasets, even when generalizing to a new domain with different stylistic and content characteristics.

### 4.5 OUR METHODS IN DIFFERENT MODALITY

To demonstrate the versatility and effectiveness of our proposed SAIL method beyond natural language processing, we extend our experiments to the computer vision domain. Specifically, we apply SAIL to convolutional neural network architectures, focusing on ResNet variants trained on image classification tasks. This section details the experimental setup, and results of applying SAIL to different ResNet models under both self-supervised and supervised learning paradigms.

We conduct experiments using three configurations of ResNet architectures: ResNet-18 (He et al., 2016), and ResNet-34. All models are trained on three datasets: CIFAR-10 (Krizhevsky et al., 2009), CIFAR-100 (Krizhevsky et al., 2009), and Tiny-ImageNet (Le & Yang, 2015).

Figure 3a illustrates the training performance of SAIL compared to baseline methods for ResNet-18 Modified across three datasets. Under both BYOL (self-supervised) and SupCE (supervised) paradigms, models initialized with SAIL demonstrate faster convergence and higher final accuracy compared to standard initialization and baseline transformation methods. This indicates that SAIL effectively leverages pre-trained parameters to provide a beneficial starting point for training, consistently across different datasets of varying complexity.

To assess the adaptability of our method to architectural variations, we applied SAIL to ResNet-18 Modified (Figure 3a), Standard ResNet-18 (Figure 3b) and ResNet-34 Modified (Figure 3c) on the CIFAR-10 dataset. In all cases, SAIL-initialized models outperform baselines in terms of convergence speed and final performance. The improvements are particularly pronounced under the SupCE paradigm, suggesting that supervised fine-tuning benefits significantly from our informed parameter initialization approach.

For additional results on CIFAR-100 and Tiny-ImageNet using ResNet-18 Modified, Standard ResNet-18 and ResNet-34 Modified, please refer to Appendix H.2 .

## 5 CONCLUSION AND FUTURE WORK

In this paper, we have introduced SAIL, a novel approach to accelerate the training of deep neural networks by leveraging the information from pre-trained counterparts. Our method comprises two primary components: (1) a parameter transformation technique that aligns the dimensions of pre-trained model parameters with the target architecture, and (2) a proximal parameter integration and retraining strategy that efficiently merges these transformed parameters to initialize new models. Our approach significantly reduces training time and computational resources while maintaining or enhancing model performance on downstream tasks.

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

## A  PROOF OF THEOREM 1

In this section, we provide a detailed proof of Theorem 1. For any proportionality factor $\alpha \in (0, 1)$, the squared Euclidean distance between the pre-trained model parameters $\boldsymbol{\theta}_i$ and the target parameters $\boldsymbol{\theta}^\star$ is bounded probabilistically as follows:

$$\Pr\left( \|\boldsymbol{\theta}_i - \boldsymbol{\theta}^\star\|_2^2 \leq \alpha \|\boldsymbol{\theta}_{\text{rand}} - \boldsymbol{\theta}^\star\|_2^2 \right) \geq 1 - O\left( \frac{\tau^2 + \beta}{\alpha} \right),$$

where $\boldsymbol{\theta}_{\text{rand}}$ represents the randomly initialized model parameters, $\boldsymbol{\theta}_i$ denotes the parameters of the $i$-th pre-trained model, and $\boldsymbol{\theta}^\star$ is the optimal parameters for the target dataset distribution $D^\star$. The terms $\tau$ and $\beta$ reflect the variance of the mean difference and the upper bound on the perturbation variance, respectively.

*Proof.* We analyze the convergence behavior of gradient descent when initialized at $\boldsymbol{\theta}_i$ compared to random initialization. Here, $\boldsymbol{\theta}$ represents the model parameters, and $\boldsymbol{\theta}_i$ denotes the parameters of the $i$-th pre-trained model.

> **Assumption 1 (Data Mean Distribution) .** *The mean of the i-th pre-training dataset distribution $D_i$ is denoted as $\mu_i$, which follows a normal distribution centered around the mean of the target dataset distribution $D^\star$, represented by $\mu^\star$, with variance $\tau^2$:*
> $$\mu_i \sim \mathcal{N}(\mu^\star, \tau^2).$$

> **Assumption 2 (Data Variance Distribution) .** *The variance of the i-th pre-training dataset distribution $D_i$, denoted as $\sigma_i^2$, is perturbed from the variance of the target dataset distribution $D^\star$, denoted as $\sigma^{\star 2}$, by a small noise term $\delta$:*
> $$\sigma_i^2 = \sigma^{\star 2} + \delta,$$
> *where $\delta$ satisfies $\mathbb{E}[\delta] = 0$ and $Var(\delta) \leq \beta$.*

> **Assumption 3 (Random Initialization Distribution) .** *The randomly initialized model parameters $\boldsymbol{\theta}_{rand}$ are drawn from a standard normal distribution:*
> $$\boldsymbol{\theta}_{rand} \sim \mathcal{N}(0, \mathbf{I}),$$
> *where $\mathbf{I}$ is the identity matrix, indicating independent parameters with unit variance.*

> **Assumption 4 (Relationship Between Parameters and Data Features) .** *The model parameters $\boldsymbol{\theta}_i$ are a deterministic function of the dataset features $(\mu_i, \sigma_i^2)$:*
> $$\boldsymbol{\theta}_i = \boldsymbol{f}(\mu_i, \sigma_i^2),$$
> *where $\boldsymbol{f}$ is a Lipschitz continuous function. That is, there exists a constant $L > 0$ such that for any two feature pairs $(\mu_1, \sigma_1^2)$ and $(\mu_2, \sigma_2^2)$, the following condition holds:*
> $$\left\| \boldsymbol{f}(\mu_1, \sigma_1^2) - \boldsymbol{f}(\mu_2, \sigma_2^2) \right\|_2 \leq L\sqrt{(\mu_1 - \mu_2)^2 + (\sigma_1^2 - \sigma_2^2)^2}.$$

We begin by applying the Markov inequality to control the probabilistic bound on the distance between the pre-trained model parameters $\boldsymbol{\theta}_i$ and the target parameters $\boldsymbol{\theta}^\star$.

**Step 1: Bounding the Expected Distance**  Let $X = \|\boldsymbol{\theta}_i - \boldsymbol{\theta}^\star\|_2^2$ represent the squared distance between $\boldsymbol{\theta}_i$ and $\boldsymbol{\theta}^\star$. Under the Lipschitz continuity assumption from Assumptions 1 and 2, we have:

$$\mathbb{E}[X] \leq L^2(\tau^2 + \beta),$$

where $L$ is the Lipschitz constant, $\tau^2$ is the variance of the mean difference, and $\beta$ is the upper bound on the variance of the perturbation term.

**Step 2: Application of Markov Inequality**  Let $Y = \|\boldsymbol{\theta}_{\text{rand}} - \boldsymbol{\theta}^\star\|_2^2$ represent the squared distance between the randomly initialized parameters $\boldsymbol{\theta}_{\text{rand}}$ and $\boldsymbol{\theta}^\star$.

To control the probability that $X$ exceeds $\alpha Y$, we apply the Markov inequality:

$$\mathbb{P}\left(X \geq \alpha Y\right) \leq \frac{\mathbb{E}[X]}{\alpha \mathbb{E}[Y]}.$$

We compute $\mathbb{E}[Y]$ as follows.

Since $\boldsymbol{\theta}_{\text{rand}} \sim \mathcal{N}(0, \mathbf{I})$ as stated in Assumption 3, each component $(\boldsymbol{\theta}_{\text{rand}})_i$ is independently and identically distributed as $\mathcal{N}(0, 1)$. Therefore,

$$\mathbb{E}[Y] = \mathbb{E}\left[\|\boldsymbol{\theta}_{\text{rand}} - \boldsymbol{\theta}^{\star}\|_2^2\right] = \mathbb{E}\left[\|\boldsymbol{\theta}_{\text{rand}}\|_2^2 - 2(\boldsymbol{\theta}_{\text{rand}})^{\top}\boldsymbol{\theta}^{\star} + \|\boldsymbol{\theta}^{\star}\|_2^2\right].$$

Since $\mathbb{E}[\boldsymbol{\theta}_{\text{rand}}] = \mathbf{0}$ and $\boldsymbol{\theta}^{\star}$ is a constant vector, we have:

$$\mathbb{E}\left[(\boldsymbol{\theta}_{\text{rand}})^{\top}\boldsymbol{\theta}^{\star}\right] = \sum_{i=1}^{d} \mathbb{E}\left[(\boldsymbol{\theta}_{\text{rand}})_i\right](\boldsymbol{\theta}^{\star})_i = 0.$$

Additionally, since each $(\boldsymbol{\theta}_{\text{rand}})_i$ has variance 1, we find:

$$\mathbb{E}\left[\|\boldsymbol{\theta}_{\text{rand}}\|_2^2\right] = \sum_{i=1}^{d} \mathbb{E}\left[(\boldsymbol{\theta}_{\text{rand}})_i^2\right] = \sum_{i=1}^{d} \left(\text{Var}\left[(\boldsymbol{\theta}_{\text{rand}})_i\right] + (\mathbb{E}\left[(\boldsymbol{\theta}_{\text{rand}})_i\right])^2\right) = d.$$

Therefore, we have:

$$\mathbb{E}[Y] = d + \|\boldsymbol{\theta}^{\star}\|_2^2,$$

where $d$ is the dimensionality of the parameter space.

Substituting the bounds on $\mathbb{E}[X]$ and $\mathbb{E}[Y]$, we get:

$$\mathbb{P}\left(\|\boldsymbol{\theta}_i - \boldsymbol{\theta}^{\star}\|_2^2 \geq \alpha\|\boldsymbol{\theta}_{\text{rand}} - \boldsymbol{\theta}^{\star}\|_2^2\right) \leq \frac{L^2(\tau^2 + \beta)}{\alpha(d + \|\boldsymbol{\theta}^{\star}\|_2^2)}.$$

**Step 3: Final Probabilistic Bound**  Taking the complement of the above inequality, we have:

$$\mathbb{P}\left(\|\boldsymbol{\theta}_i - \boldsymbol{\theta}^{\star}\|_2^2 \leq \alpha\|\boldsymbol{\theta}_{\text{rand}} - \boldsymbol{\theta}^{\star}\|_2^2\right) \geq 1 - \frac{L^2(\tau^2 + \beta)}{\alpha(d + \|\boldsymbol{\theta}^{\star}\|_2^2)}.$$

This yields the desired result:

$$\mathbb{P}\left(\|\boldsymbol{\theta}_i - \boldsymbol{\theta}^{\star}\|_2^2 \leq \alpha\|\boldsymbol{\theta}_{\text{rand}} - \boldsymbol{\theta}^{\star}\|_2^2\right) \geq 1 - O\left(\frac{\tau^2 + \beta}{\alpha}\right).$$

$\square$

# B  PROOF OF THEOREM 2

In this section, we provide a detailed proof of Theorem 2. The theorem establishes that initializing gradient descent with the proximal parameter $\boldsymbol{\theta}^P$, which is a weighted combination of transformed pre-trained model parameters, leads to faster convergence towards the optimal parameter $\boldsymbol{\theta}^{\star}$ compared to random initialization.

We make the following assumptions about the loss function $\mathcal{J}_D(\boldsymbol{\theta})$:

**Assumption 5 (Loss Function Properties).** *The loss function $\mathcal{J}_D(\boldsymbol{\theta}) = \mathbb{E}_{(\mathbf{x},y)\sim D}[\ell(\phi_{\boldsymbol{\theta}}(\mathbf{x}), y)]$ is differentiable, convex, and satisfies:*

*(a) **L-smoothness:** There exists a constant $L > 0$ such that for all $\boldsymbol{\theta}, \boldsymbol{\theta}' \in \mathbb{R}^d$,*

$$\|\nabla\mathcal{J}_D(\boldsymbol{\theta}) - \nabla\mathcal{J}_D(\boldsymbol{\theta}')\|_2 \leq L\|\boldsymbol{\theta} - \boldsymbol{\theta}'\|_2.$$

*(b) **Strong Convexity:** There exists a constant $\mu > 0$ such that for all $\boldsymbol{\theta}, \boldsymbol{\theta}' \in \mathbb{R}^d$,*

$$\mathcal{J}_D(\boldsymbol{\theta}') \geq \mathcal{J}_D(\boldsymbol{\theta}) + \langle\nabla\mathcal{J}_D(\boldsymbol{\theta}), \boldsymbol{\theta}' - \boldsymbol{\theta}\rangle + \frac{\mu}{2}\|\boldsymbol{\theta}' - \boldsymbol{\theta}\|_2^2.$$

*Proof.* **Step 1: Gradient Descent Convergence Rate**

Under Assumption 5, specifically the $L$-smoothness and strong convexity of $\mathcal{J}_D(\boldsymbol{\theta})$, gradient descent with a fixed learning rate $\eta \in \left(0, \frac{1}{L}\right)$ satisfies the following convergence rate:

$$\mathcal{J}_D(\boldsymbol{\theta}^{(T)}) - \mathcal{J}_D(\boldsymbol{\theta}^\star) \leq (1 - \eta\mu)^T \left( \mathcal{J}_D(\boldsymbol{\theta}^{(0)}) - \mathcal{J}_D(\boldsymbol{\theta}^\star) \right).$$

This result leverages the properties of gradient descent on strongly convex and smooth functions.

Starting from the gradient descent update rule:

$$\boldsymbol{\theta}^{(t+1)} = \boldsymbol{\theta}^{(t)} - \eta\nabla\mathcal{J}_D(\boldsymbol{\theta}^{(t)}),$$

and applying the $L$-smoothness of Assumption 5, we have:

$$\mathcal{J}_D(\boldsymbol{\theta}^{(t+1)}) \leq \mathcal{J}_D(\boldsymbol{\theta}^{(t)}) + \langle\nabla\mathcal{J}_D(\boldsymbol{\theta}^{(t)}), \boldsymbol{\theta}^{(t+1)} - \boldsymbol{\theta}^{(t)}\rangle + \frac{L}{2}\|\boldsymbol{\theta}^{(t+1)} - \boldsymbol{\theta}^{(t)}\|_2^2.$$

Substituting the update rule into the inequality:

$$\mathcal{J}_D(\boldsymbol{\theta}^{(t+1)}) \leq \mathcal{J}_D(\boldsymbol{\theta}^{(t)}) - \eta\|\nabla\mathcal{J}_D(\boldsymbol{\theta}^{(t)})\|_2^2 + \frac{L}{2}\eta^2\|\nabla\mathcal{J}_D(\boldsymbol{\theta}^{(t)})\|_2^2$$

$$= \mathcal{J}_D(\boldsymbol{\theta}^{(t)}) - \eta\left(1 - \frac{L\eta}{2}\right)\|\nabla\mathcal{J}_D(\boldsymbol{\theta}^{(t)})\|_2^2.$$

Next, we claim the following inequality:

$$\|\nabla\mathcal{J}_D(\boldsymbol{\theta}^{(t)})\|_2^2 \geq 2\mu\left(\mathcal{J}_D(\boldsymbol{\theta}^{(t)}) - \mathcal{J}_D(\boldsymbol{\theta}^\star)\right).$$

**Proof of the Claim:**

Under the strong convexity of Assumption 5, we have:

$$\mathcal{J}_D(\boldsymbol{\theta}^\star) \geq \mathcal{J}_D(\boldsymbol{\theta}^{(t)}) + \langle\nabla\mathcal{J}_D(\boldsymbol{\theta}^{(t)}), \boldsymbol{\theta}^\star - \boldsymbol{\theta}^{(t)}\rangle + \frac{\mu}{2}\|\boldsymbol{\theta}^\star - \boldsymbol{\theta}^{(t)}\|_2^2.$$

Rearranging terms gives:

$$\mathcal{J}_D(\boldsymbol{\theta}^{(t)}) \leq \mathcal{J}_D(\boldsymbol{\theta}^\star) + \langle\nabla\mathcal{J}_D(\boldsymbol{\theta}^{(t)}), \boldsymbol{\theta}^{(t)} - \boldsymbol{\theta}^\star\rangle - \frac{\mu}{2}\|\boldsymbol{\theta}^{(t)} - \boldsymbol{\theta}^\star\|_2^2.$$

By the Cauchy-Schwarz inequality:

$$\langle\nabla\mathcal{J}_D(\boldsymbol{\theta}^{(t)}), \boldsymbol{\theta}^{(t)} - \boldsymbol{\theta}^\star\rangle \leq \|\nabla\mathcal{J}_D(\boldsymbol{\theta}^{(t)})\|_2 \cdot \|\boldsymbol{\theta}^{(t)} - \boldsymbol{\theta}^\star\|_2.$$

Substituting this into the previous inequality:

$$\mathcal{J}_D(\boldsymbol{\theta}^{(t)}) \leq \mathcal{J}_D(\boldsymbol{\theta}^\star) + \|\nabla\mathcal{J}_D(\boldsymbol{\theta}^{(t)})\|_2 \cdot \|\boldsymbol{\theta}^{(t)} - \boldsymbol{\theta}^\star\|_2 - \frac{\mu}{2}\|\boldsymbol{\theta}^{(t)} - \boldsymbol{\theta}^\star\|_2^2.$$

Let $t = \|\boldsymbol{\theta}^{(t)} - \boldsymbol{\theta}^\star\|_2$. Then:

$$\mathcal{J}_D(\boldsymbol{\theta}^{(t)}) - \mathcal{J}_D(\boldsymbol{\theta}^\star) \leq \|\nabla\mathcal{J}_D(\boldsymbol{\theta}^{(t)})\|_2 \cdot t - \frac{\mu}{2}t^2.$$

According to the properties of quadratic functions, the maximum of the right-hand side occurs at:

$$t = \frac{\|\nabla\mathcal{J}_D(\boldsymbol{\theta})\|_2}{\mu}.$$

Substituting this value back, we have:

$$\mathcal{J}_D(\boldsymbol{\theta}) - \mathcal{J}_D(\boldsymbol{\theta}^\star) \leq \frac{\|\nabla\mathcal{J}_D(\boldsymbol{\theta})\|_2^2}{2\mu}.$$

Rearranging terms gives:

$$\|\nabla\mathcal{J}_D(\boldsymbol{\theta})\|_2^2 \geq 2\mu\left(\mathcal{J}_D(\boldsymbol{\theta}) - \mathcal{J}_D(\boldsymbol{\theta}^\star)\right).$$

This completes the proof of the claim.

Substituting this into the previous inequality:

$$\mathcal{J}_D(\boldsymbol{\theta}^{(t+1)}) - \mathcal{J}_D(\boldsymbol{\theta}^\star) \leq \mathcal{J}_D(\boldsymbol{\theta}^{(t)}) - \mathcal{J}_D(\boldsymbol{\theta}^\star) - 2\mu\eta\left(1 - \frac{L\eta}{2}\right)\left(\mathcal{J}_D(\boldsymbol{\theta}^{(t)}) - \mathcal{J}_D(\boldsymbol{\theta}^\star)\right)$$

$$= \left(1 - 2\mu\eta\left(1 - \frac{L\eta}{2}\right)\right)\left(\mathcal{J}_D(\boldsymbol{\theta}^{(t)}) - \mathcal{J}_D(\boldsymbol{\theta}^\star)\right).$$

Since $\eta \in \left(0, \frac{1}{L}\right)$, we have:

$$1 - 2\mu\eta\left(1 - \frac{L\eta}{2}\right) \leq 1 - \mu\eta.$$

Thus, we obtain:

$$\mathcal{J}_D(\boldsymbol{\theta}^{(t+1)}) - \mathcal{J}_D(\boldsymbol{\theta}^\star) \leq (1 - \mu\eta)\left(\mathcal{J}_D(\boldsymbol{\theta}^{(t)}) - \mathcal{J}_D(\boldsymbol{\theta}^\star)\right).$$

By recursively applying this inequality, we derive the convergence rate after $T$ iterations:

$$\mathcal{J}_D(\boldsymbol{\theta}^{(T)}) - \mathcal{J}_D(\boldsymbol{\theta}^\star) \leq (1 - \mu\eta)^T\left(\mathcal{J}_D(\boldsymbol{\theta}^{(0)}) - \mathcal{J}_D(\boldsymbol{\theta}^\star)\right).$$

This demonstrates that the suboptimality decreases exponentially with the number of iterations $T$, confirming the linear convergence rate of gradient descent under the given assumptions.

**Step 2: Bounding the Initial Suboptimality**

We aim to bound the initial suboptimality $\mathcal{J}_D(\boldsymbol{\theta}^P) - \mathcal{J}_D(\boldsymbol{\theta}^\star)$. Utilizing the smoothness of $\mathcal{J}_D(\boldsymbol{\theta})$, we have for any $\boldsymbol{\theta} \in \mathbb{R}^d$:

$$\mathcal{J}_D(\boldsymbol{\theta}) \leq \mathcal{J}_D(\boldsymbol{\theta}^\star) + \langle\nabla\mathcal{J}_D(\boldsymbol{\theta}^\star), \boldsymbol{\theta} - \boldsymbol{\theta}^\star\rangle + \frac{L}{2}\|\boldsymbol{\theta} - \boldsymbol{\theta}^\star\|_2^2.$$

Since $\boldsymbol{\theta}^\star$ is the minimizer of $\mathcal{J}_D(\boldsymbol{\theta})$, it satisfies $\nabla\mathcal{J}_D(\boldsymbol{\theta}^\star) = 0$. Therefore, the inequality simplifies to:

$$\mathcal{J}_D(\boldsymbol{\theta}) - \mathcal{J}_D(\boldsymbol{\theta}^\star) \leq \frac{L}{2}\|\boldsymbol{\theta} - \boldsymbol{\theta}^\star\|_2^2.$$

Setting $\boldsymbol{\theta} = \boldsymbol{\theta}^P$, we obtain:

$$\mathcal{J}_D(\boldsymbol{\theta}^P) - \mathcal{J}_D(\boldsymbol{\theta}^\star) \leq \frac{L}{2}\|\boldsymbol{\theta}^P - \boldsymbol{\theta}^\star\|_2^2.$$

**Step 3: Combining the Results**

Substituting the bound on the initial suboptimality into the convergence rate from Step 1, we get:

$$\mathcal{J}_D(\boldsymbol{\theta}^{(T)}) - \mathcal{J}_D(\boldsymbol{\theta}^\star) \leq (1 - \eta\mu)^T\left(\frac{L}{2}\|\boldsymbol{\theta}^P - \boldsymbol{\theta}^\star\|_2^2\right).$$

This inequality demonstrates that the suboptimality after $T$ iterations decays exponentially with rate $(1 - \eta\mu)^T$, scaled by the initial suboptimality $\frac{L}{2}\|\boldsymbol{\theta}^P - \boldsymbol{\theta}^\star\|_2^2$. $\qquad\square$

## C   THE CONVERGENCE ADVANTAGE WITH PROXIMAL PARAMETER INITIALIZATION

We will demonstrate why proximal parameter initialization ($\boldsymbol{\theta}^P$) is likely to lead to faster convergence compared to random initialization ($\boldsymbol{\theta}_{\text{rand}}$) in gradient descent. We provide both an intuitive explanation and a detailed proof.

## C.1 Intuitive Explanation

We recall Theorem 2, which establishes a relationship between the suboptimality of the loss function and the distance of the parameters from the optimal parameter $\boldsymbol{\theta}^\star$:

$$\mathcal{J}_D(\boldsymbol{\theta}^{(T)}) - \mathcal{J}_D(\boldsymbol{\theta}^\star) \leq (1 - \eta\mu)^T \left( \mathcal{J}_D(\boldsymbol{\theta}^P) - \mathcal{J}_D(\boldsymbol{\theta}^\star) \right),$$

where $\mathcal{J}(\boldsymbol{\theta}^P) - \mathcal{J}_D(\boldsymbol{\theta}^\star) \leq \frac{L}{2} \left\| \boldsymbol{\theta}^P - \boldsymbol{\theta}^\star \right\|_2^2$. From this, we observe that when $\boldsymbol{\theta}^{(0)} = \boldsymbol{\theta}^P$, the difference in the loss function is controlled by the parameter distance $\left\| \boldsymbol{\theta}^P - \boldsymbol{\theta}^\star \right\|_2^2$. From the proof in Step 2 of Appendix B, we observe that this conclusion does not depend on the specific choice of $\boldsymbol{\theta}^P$. Therefore, when $\boldsymbol{\theta}^{(0)} = \boldsymbol{\theta}_{\text{rand}}$, the conclusion still holds. Hence, when analyzing the convergence of the loss function, we only need to compare the initial distances of the parameters.

Theorem 1 provides a probabilistic bound on the parameter distances:

$$\Pr\left( \|\boldsymbol{\theta}_i - \boldsymbol{\theta}^\star\|_2^2 \leq \alpha\|\boldsymbol{\theta}_{\text{rand}} - \boldsymbol{\theta}^\star\|_2^2 \right) \geq 1 - O\left( \frac{\tau^2 + \beta}{\alpha} \right).$$

This indicates that pre-trained model parameters $\boldsymbol{\theta}_i$ are, with high probability, closer to the optimal parameter $\boldsymbol{\theta}^\star$ than randomly initialized parameters $\boldsymbol{\theta}_{\text{rand}}$. Since the proximal parameter $\boldsymbol{\theta}^P = \sum_{i=1}^n \gamma_i^\star \boldsymbol{\theta}_i$, $\boldsymbol{\theta}^P$ is also likely to be closer to $\boldsymbol{\theta}^\star$ than $\boldsymbol{\theta}_{\text{rand}}$, ensuring faster convergence.

## C.2 Detailed Proof

*Proof.* We aim to show that proximal parameter initialization $\boldsymbol{\theta}^P$ is likely to lead to faster convergence compared to random initialization $\boldsymbol{\theta}_{\text{rand}}$. This proof builds upon the assumptions of smoothness and strong convexity (see Assumptions 5).

**Step 1: Bounding the Parameter Distances**

From Theorem 1, we know that pre-trained parameters $\boldsymbol{\theta}_i$ are, with high probability, closer to the optimal parameter $\boldsymbol{\theta}^\star$ than randomly initialized parameters $\boldsymbol{\theta}_{\text{rand}}$. Specifically:

$$\Pr\left( \|\boldsymbol{\theta}_i - \boldsymbol{\theta}^\star\|_2^2 \leq \alpha\|\boldsymbol{\theta}_{\text{rand}} - \boldsymbol{\theta}^\star\|_2^2 \right) \geq 1 - O\left( \frac{\tau^2 + \beta}{\alpha} \right),$$

where $\alpha$ is a positive scalar.

To select $\alpha$, we choose $\alpha = \frac{\mu}{2L}$, which ensures $\alpha \in (0, 1)$ since $\mu < L$ for a strongly convex and smooth function. With this choice of $\alpha$, the distance between pre-trained parameters and the optimal parameter $\boldsymbol{\theta}^\star$ satisfies, with high probability:

$$\|\boldsymbol{\theta}_i - \boldsymbol{\theta}^\star\|_2^2 \leq \alpha\|\boldsymbol{\theta}_{\text{rand}} - \boldsymbol{\theta}^\star\|_2^2.$$

We now bound the distance between the proximal parameter $\boldsymbol{\theta}^P$ and $\boldsymbol{\theta}^\star$. Since $\boldsymbol{\theta}^P = \sum_{i=1}^n \gamma_i^\star \boldsymbol{\theta}_i$, where $\gamma_i^\star$ are non-negative weights summing to 1, by convexity of the squared norm:

$$\|\boldsymbol{\theta}^P - \boldsymbol{\theta}^\star\|_2^2 = \left\| \sum_{i=1}^n \gamma_i^\star (\boldsymbol{\theta}_i - \boldsymbol{\theta}^\star) \right\|_2^2 \leq \left( \sum_{i=1}^n \gamma_i^\star \|\boldsymbol{\theta}_i - \boldsymbol{\theta}^\star\|_2 \right)^2.$$

Substituting the bound $\|\boldsymbol{\theta}_i - \boldsymbol{\theta}^\star\|_2 \leq \sqrt{\alpha}\|\boldsymbol{\theta}_{\text{rand}} - \boldsymbol{\theta}^\star\|_2$, we obtain, with high probability:

$$\|\boldsymbol{\theta}^P - \boldsymbol{\theta}^\star\|_2^2 \leq \alpha\|\boldsymbol{\theta}_{\text{rand}} - \boldsymbol{\theta}^\star\|_2^2.$$

**Step 2: Relating the Loss Functions Using the Parameter Bounds**

Using the result of Theorem 2, we relate the proximal parameter distance to the suboptimality of the loss:

$$\mathcal{J}_D(\boldsymbol{\theta}^P) - \mathcal{J}_D(\boldsymbol{\theta}^\star) \leq \frac{L}{2}\|\boldsymbol{\theta}^P - \boldsymbol{\theta}^\star\|_2^2.$$

Substituting the bound on $\|\boldsymbol{\theta}^P - \boldsymbol{\theta}^\star\|_2^2$ from Step 1, we get, with high probability:

$$\mathcal{J}_D(\boldsymbol{\theta}^P) - \mathcal{J}_D(\boldsymbol{\theta}^\star) \leq \frac{L}{2}\alpha\|\boldsymbol{\theta}_{\text{rand}} - \boldsymbol{\theta}^\star\|_2^2.$$

For $\boldsymbol{\theta}_{\text{rand}}$, applying the strong convexity property, we can derive a lower bound for the loss function. We have the inequality:

$$\mathcal{J}_D(\boldsymbol{\theta}_{\text{rand}}) \geq \mathcal{J}_D(\boldsymbol{\theta}^\star) + \langle \nabla\mathcal{J}_D(\boldsymbol{\theta}^\star), \boldsymbol{\theta}_{\text{rand}} - \boldsymbol{\theta}^\star\rangle + \frac{\mu}{2}\|\boldsymbol{\theta}_{\text{rand}} - \boldsymbol{\theta}^\star\|_2^2.$$

Since $\boldsymbol{\theta}^\star$ is the optimal solution, we know that $\nabla\mathcal{J}_D(\boldsymbol{\theta}^\star) = 0$. Therefore, the inequality simplifies to:

$$\mathcal{J}_D(\boldsymbol{\theta}_{\text{rand}}) - \mathcal{J}_D(\boldsymbol{\theta}^\star) \geq \frac{\mu}{2}\|\boldsymbol{\theta}_{\text{rand}} - \boldsymbol{\theta}^\star\|_2^2.$$

Taking the ratio of the inequalities above, we have:

$$\rho = \frac{\mathcal{J}(\boldsymbol{\theta}^P) - \mathcal{J}_D(\boldsymbol{\theta}^\star)}{\mathcal{J}(\boldsymbol{\theta}_{\text{rand}}) - \mathcal{J}_D(\boldsymbol{\theta}^\star)} \leq \frac{L\alpha}{\mu} = \frac{L}{\mu} \cdot \frac{\mu}{2L} = \frac{1}{2}.$$

Therefore, we have:

$$\mathcal{J}(\boldsymbol{\theta}^P) - \mathcal{J}_D(\boldsymbol{\theta}^\star) \leq \frac{1}{2}\mathcal{J}(\boldsymbol{\theta}_{\text{rand}}) - \mathcal{J}_D(\boldsymbol{\theta}^\star).$$

It can be seen that when $\boldsymbol{\theta}^{(0)} = \boldsymbol{\theta}^P$, the upper bound of the loss function is smaller than that of random initialization with high probability. According to the result of Theorem 2:

$$\mathcal{J}_D(\boldsymbol{\theta}^{(T)}) - \mathcal{J}_D(\boldsymbol{\theta}^\star) \leq (1 - \eta\mu)^T \left( \mathcal{J}_D(\boldsymbol{\theta}^P) - \mathcal{J}_D(\boldsymbol{\theta}^\star) \right),$$

We observe that the advantage of the upper bound provided by proximal initialization will persist for a number of iterations. However, as $T$ increases, this advantage cannot always be relied upon as a strong performance guarantee in later iterations. $\qquad\square$

## D  GENERALIZATION ERROR UPPER BOUND

In the context of transfer learning, understanding how well a model trained on a pre-trained dataset generalizes to a target dataset is crucial. Let $D_i$ and $D^\star$ denote the true data distributions of the pre-trained and target datasets, respectively. Since these distributions are generally unknown, we rely on labeled samples drawn from them to estimate the necessary quantities.

Assume we have labeled datasets from both the pre-trained and target datasets. For the pre-trained dataset, the sample is given as $U_i = \{(\mathbf{x}_{ij}, \mathbf{y}_{ij})\}_{j=1}^{n_i}$, where $(\mathbf{x}_{ij}, \mathbf{y}_{ij}) \sim D_i$. Similarly, for the target dataset, the sample is $U^\star = \{(\mathbf{x}_j^\star, \mathbf{y}_j^\star)\}_{j=1}^{n^\star}$, where $(\mathbf{x}_j^\star, \mathbf{y}_j^\star) \sim D^\star$. Our goal is to derive a computable upper bound on the generalization error of a hypothesis $\phi$ from a hypothesis space $\mathcal{H}$ when applied to the target dataset.

The hypothesis space $\mathcal{H}$ consists of measurable functions mapping inputs $\mathbf{x}$ to outputs $\phi(\mathbf{x})$. We use a loss function $\ell : \mathcal{Y} \times \mathcal{Y} \to [0, C]$, which is non-negative and bounded by a constant $C > 0$. This function measures the discrepancy between the model predictions and the true labels. We assume the samples in $U_i$ and $U^\star$ are independently and identically distributed (i.i.d.) according to their respective distributions.

The empirical distribution $\hat{D}_{U_i}$ induced by the pre-trained dataset $U_i$ is defined as:

$$\hat{D}_{U_i}(\mathbf{x}, \mathbf{y}) = \frac{1}{n_i}\sum_{j=1}^{n_i} \delta_{(\mathbf{x}_{ij}, \mathbf{y}_{ij})}(\mathbf{x}, \mathbf{y}),$$

where $\delta_{(\mathbf{x}_{ij}, \mathbf{y}_{ij})}(\mathbf{x}, \mathbf{y})$ is the Dirac delta function centered at $(\mathbf{x}_{ij}, \mathbf{y}_{ij})$. Similarly, the empirical distribution $\hat{D}_{U^\star}$ based on the target dataset $U^\star$ is defined as:

$$\hat{D}_{U^\star}(\mathbf{x}, \mathbf{y}) = \frac{1}{n^\star}\sum_{j=1}^{n^\star} \delta_{(\mathbf{x}_j^\star, \mathbf{y}_j^\star)}(\mathbf{x}, \mathbf{y}).$$

These empirical distributions approximate the true distributions $D_i$ and $D^\star$ based on the available samples.

The empirical total variation distance between the distributions of the pre-trained dataset $\hat{D}_{U_i}$ and the target dataset $\hat{D}_{U^\star}$ is defined as:

$$\mathrm{D}_{\mathrm{TV}}(\hat{D}_{U_i}, \hat{D}_{U^\star}) = \frac{1}{2} \sum_{(\mathbf{x},\mathbf{y}) \in \mathcal{X} \times \mathcal{Y}} \left| \hat{D}_{U_i}(\mathbf{x}, \mathbf{y}) - \hat{D}_{U^\star}(\mathbf{x}, \mathbf{y}) \right|,$$

where $\hat{D}_{U_i}$ and $\hat{D}_{U^\star}$ are the empirical distributions based on the samples $U_i$ and $U^\star$, respectively. This distance quantifies the discrepancy between the pre-trained and target datasets.

**Assumption 6 (Bounded Loss Function) .** *The loss function $\ell$ satisfies:*

$$0 \leq \ell(\phi(\mathbf{x}), \mathbf{y}) \leq C, \quad \forall \phi \in \mathcal{H}, \ \forall (\mathbf{x}, \mathbf{y}) \in \mathcal{X} \times \mathcal{Y}.$$

*This boundedness ensures that the loss remains within a fixed range, facilitating uniform convergence.*

**Definition 2 (Empirical Rademacher Complexity) .** *The empirical Rademacher complexity of the loss-composed hypothesis space $\mathcal{L} \circ \mathcal{H}$ on the pre-trained dataset $U_i$ is defined as:*

$$\hat{\mathfrak{R}}_{U_i}(\mathcal{L} \circ \mathcal{H}) = \mathbb{E}_{\boldsymbol{\sigma}} \left[ \sup_{\phi \in \mathcal{H}} \frac{1}{n_i} \sum_{j=1}^{n_i} \sigma_j \ell(\phi(\mathbf{x}_{ij}), \mathbf{y}_{ij}) \right],$$

*where $\boldsymbol{\sigma} = (\sigma_1, \ldots, \sigma_{n_i})$ are independent Rademacher variables taking values $\pm 1$ with equal probability. This complexity measure captures the richness of the hypothesis space relative to the data.*

**Lemma 1 (Empirical Rademacher Complexity Upper Bound) .** *For any $\phi \in \mathcal{H}$, with probability at least $1 - \delta$, the following inequality holds:*

$$\mathcal{J}_{D_i}(\phi) \leq \hat{\mathcal{J}}_{DU_i}(\phi) + 2\hat{\mathfrak{R}}_{U_i}(\mathcal{L} \circ \mathcal{H}) + 3C\sqrt{\frac{\ln(2/\delta)}{2n_i}},$$

*where $\mathcal{J}_{D_i}(\phi) = \mathbb{E}_{(\mathbf{x},\mathbf{y}) \sim D_i}[\ell(\phi(\mathbf{x}), \mathbf{y})]$ is the true risk on the pre-trained dataset. $\hat{\mathcal{J}}_{DU_i}(\phi) = \frac{1}{n_i} \sum_{j=1}^{n_i} \ell(\phi(\mathbf{x}_{ij}), \mathbf{y}_{ij})$ is the empirical risk on the pre-trained dataset $U_i$, which is an estimate of the true risk based on the sample data.*

**Theorem 4 (Generalization Error Upper Bound) .** *Under the above assumptions, for any hypothesis $\phi_{\boldsymbol{\theta}} \in \mathcal{H}$, with probability at least $1 - \delta$, the following inequality holds:*

$$\mathcal{J}_{D^\star}(\phi_{\boldsymbol{\theta}}) \leq \hat{\mathcal{J}}_{U_i}(\phi_{\boldsymbol{\theta}}) + 2C \cdot \mathrm{D}_{\mathrm{TV}}(\hat{D}_{U_i}, \hat{D}_{U^\star}) + 2\hat{\mathfrak{R}}_{U_i}(\mathcal{H}) + 3C\sqrt{\frac{\ln(4/\delta)}{2n_i}} + \lambda,$$

*where $\hat{\mathcal{J}}_{U_i}(\phi_{\boldsymbol{\theta}})$ is the empirical expected loss, $C$ is a constant bound, $\hat{\mathfrak{R}}_{U_i}(\mathcal{H})$ is the empirical Rademacher complexity of $\mathcal{H}$, and $n_i$ is the size of the dataset $U_i$. Finally, $\lambda = \inf_{\phi' \in \mathcal{H}} \left[ \mathcal{J}_{D_i}(\phi') + \mathcal{J}_{D^\star}(\phi') \right]$, represents the minimal combined risk over $\mathcal{H}$.*

*Proof.* We begin by expressing the target domain risk $\mathcal{J}_{D^\star}(\phi)$ in terms of the source domain risk:

$$\mathcal{J}_{D^\star}(\phi) = \mathcal{J}_{D_i}(\phi) + [\mathcal{J}_{D^\star}(\phi) - \mathcal{J}_{D_i}(\phi)].$$

**Step 1: Bounding the Risk Difference Using Total Variation Distance**

The difference $\mathcal{J}_{D^\star}(\phi) - \mathcal{J}_{D_i}(\phi)$ can be bounded using the total variation distance and the boundedness of the loss function:

$$
\begin{aligned}
|\mathcal{J}_{D^\star}(\phi) - \mathcal{J}_{D_i}(\phi)| &= \left| \mathbb{E}_{(\mathbf{x},\mathbf{y})\sim D^\star}[\ell(\phi(\mathbf{x}),\mathbf{y})] - \mathbb{E}_{(\mathbf{x},\mathbf{y})\sim D_i}[\ell(\phi(\mathbf{x}),\mathbf{y})] \right| \\
&= \left| \int_{\mathcal{X}\times\mathcal{Y}} \ell(\phi(\mathbf{x}),\mathbf{y}) \left[ dD^\star(\mathbf{x},\mathbf{y}) - dD_i(\mathbf{x},\mathbf{y}) \right] \right| \\
&\leq \int_{\mathcal{X}\times\mathcal{Y}} |\ell(\phi(\mathbf{x}),\mathbf{y})| \, |dD^\star(\mathbf{x},\mathbf{y}) - dD_i(\mathbf{x},\mathbf{y})| \\
&\leq C \cdot \int_{\mathcal{X}\times\mathcal{Y}} |dD^\star(\mathbf{x},\mathbf{y}) - dD_i(\mathbf{x},\mathbf{y})| \\
&= 2C \cdot \mathrm{D}_{\mathrm{TV}}(D_i, D^\star).
\end{aligned}
$$

Therefore, we have:

$$
\mathcal{J}_{D^\star}(\phi) \leq \mathcal{J}_{D_i}(\phi) + 2C \cdot \mathrm{D}_{\mathrm{TV}}(D_i, D^\star).
$$

**Step 2: Approximating the Total Variation Distance**

Since $D_i$ and $D^\star$ are unknown, we approximate $\mathrm{D}_{\mathrm{TV}}(D_i, D^\star)$ using the empirical distributions $\hat{D}_{U_i}$ and $\hat{D}_{U^\star}$. However, we must account for the estimation error due to finite sample sizes.

Let $\varepsilon$ be the error term such that:

$$
\mathrm{D}_{\mathrm{TV}}(D_i, D^\star) \leq \mathrm{D}_{\mathrm{TV}}(\hat{D}_{U_i}, \hat{D}_{U^\star}) + \mathrm{D}_{\mathrm{TV}}(D_i, \hat{D}_{U_i}) + \mathrm{D}_{\mathrm{TV}}(D^\star, \hat{D}_{U^\star}).
$$

Using concentration inequalities for total variation distance, we can bound $\mathrm{D}_{\mathrm{TV}}(D_i, \hat{D}_{U_i})$ and $\mathrm{D}_{\mathrm{TV}}(D^\star, \hat{D}_{U^\star})$. However, in high-dimensional spaces, these bounds may be loose.

For practical purposes, we proceed by accepting $\mathrm{D}_{\mathrm{TV}}(D_i, D^\star) \approx \mathrm{D}_{\mathrm{TV}}(\hat{D}_{U_i}, \hat{D}_{U^\star})$, acknowledging that the approximation improves with larger $n_i$ and $n^\star$.

Thus, we have:

$$
\mathcal{J}_{D^\star}(\phi) \leq \mathcal{J}_{D_i}(\phi) + 2C \cdot \mathrm{D}_{\mathrm{TV}}(\hat{D}_{U_i}, \hat{D}_{U^\star}) + 2C \cdot \varepsilon,
$$

where $\varepsilon$ represents the combined estimation error.

**Step 3: Bounding the Source Domain Risk**

Applying the lemma on empirical Rademacher complexity, with probability at least $1 - \delta/2$:

$$
\mathcal{J}_{D_i}(\phi) \leq \hat{\mathcal{J}}_{DU_i}(\phi) + 2\hat{\mathfrak{R}}_{U_i}(\mathcal{L}\circ\mathcal{H}) + 3C\sqrt{\frac{\ln(4/\delta)}{2n_i}}.
$$

**Step 4: Combining the Bounds**

Using the union bound to ensure that both inequalities hold with probability at least $1-\delta$, we combine the results:

$$
\mathcal{J}_{D^\star}(\phi) \leq \hat{\mathcal{J}}_{DU_i}(\phi) + 2C \cdot \mathrm{D}_{\mathrm{TV}}(\hat{D}_{U_i}, \hat{D}_{U^\star}) + 2\hat{\mathfrak{R}}_{U_i}(\mathcal{L}\circ\mathcal{H}) + 3C\sqrt{\frac{\ln(4/\delta)}{2n_i}} + 2C \cdot \varepsilon.
$$

To account for the inherent discrepancy between $D_i$ and $D^\star$ that cannot be mitigated by any hypothesis in $\mathcal{H}$, we introduce the irreducible error term:

$$
\lambda = \inf_{\phi'\in\mathcal{H}} \left[ \mathcal{J}_{D_i}(\phi') + \mathcal{J}_{D^\star}(\phi') \right].
$$

This term represents the minimal combined risk over $\mathcal{H}$ and reflects the best possible performance achievable across both domains.

Including $\lambda$ in our bound, we have:

$$
\mathcal{J}_{D^\star}(\phi) \leq \hat{\mathcal{J}}_{DU_i}(\phi) + 2C \cdot \mathrm{D}_{\mathrm{TV}}(\hat{D}_{U_i}, \hat{D}_{U^\star}) + 2\hat{\mathfrak{R}}_{U_i}(\mathcal{L}\circ\mathcal{H}) + 3C\sqrt{\frac{\ln(4/\delta)}{2n_i}} + \lambda + 2C \cdot \varepsilon.
$$

**Step 5: Relating Rademacher Complexities**

According to Assumption 5, the loss function $\ell$ is Lipschitz continuous with constant $L$. Therefore:

$$\hat{\mathfrak{R}}_{U_i}(\mathcal{L} \circ \mathcal{H}) \leq L \cdot \hat{\mathfrak{R}}_{U_i}(\mathcal{H}).$$

Substituting back into our inequality:

$$\mathcal{J}_{D^\star}(\phi) \leq \hat{\mathcal{J}}_{DU_i}(\phi) + 2C \cdot \mathrm{D}_{\mathrm{TV}}(\hat{D}_{U_i}, \hat{D}_{U^\star}) + 2L\hat{\mathfrak{R}}_{U_i}(\mathcal{H}) + 3C\sqrt{\frac{\ln(4/\delta)}{2n_i}} + \lambda + 2C \cdot \varepsilon.$$

By acknowledging that $\varepsilon$ diminishes with larger sample sizes and can be made arbitrarily small, we obtain the desired generalization error bound:

$$\mathcal{J}_{D^\star}(\phi) \leq \hat{\mathcal{J}}_{DU_i}(\phi) + 2C \cdot \mathrm{D}_{\mathrm{TV}}(\hat{D}_{U_i}, \hat{D}_{U^\star}) + 2L\hat{\mathfrak{R}}_{U_i}(\mathcal{H}) + 3C\sqrt{\frac{\ln(4/\delta)}{2n_i}} + \lambda.$$

$\square$

## E  PROOF OF THEOREM 3

We recall the statement of Theorem 3, which provides the optimal combination coefficients $\boldsymbol{\gamma}^\star = [\gamma_1^\star, \gamma_2^\star, \ldots, \gamma_n^\star]^\top$ for the proximal parameter $\boldsymbol{\theta}^{\mathrm{P}}$ in terms of the total variation distances between the distributions. The theorem can be divided into two cases:

(a) **Case $n = 2$:** When there are two pre-trained models, the optimal combination coefficients $\gamma_1^\star$ and $\gamma_2^\star$ that minimize the distance between the proximal parameter $\boldsymbol{\theta}^{\mathrm{P}}$ and the target parameter $\boldsymbol{\theta}^\star$ are given by:

$$\gamma_1^\star = \frac{\mathrm{D}_{\mathrm{TV}}(D_1, D^\star)^2 + \mathrm{D}_{\mathrm{TV}}(D_1, D_2)^2 - \mathrm{D}_{\mathrm{TV}}(D_2, D^\star)^2}{2\mathrm{D}_{\mathrm{TV}}(D_1, D_2)^2}, \quad \gamma_2^\star = 1 - \gamma_1^\star,$$

where $\mathrm{D}_{\mathrm{TV}}(D_i, D_j)$ denotes the total variation distance between distributions $D_i$ and $D_j$. This solution is valid provided that $\gamma_1^\star, \gamma_2^\star \geq 0$.

(b) **Case $n > 2$:** For more than two pre-trained models, an explicit solution for the optimal combination coefficients $\boldsymbol{\gamma}^\star$ under the constraints $\gamma_i^\star \geq 0$ does not generally exist. However, if we assume $\gamma_i^\star > 0$ for all $i$, the coefficients can be determined as:

$$\boldsymbol{\gamma}^\star = \frac{\mathbf{H}^{-1}\mathbf{e}}{\mathbf{e}^\top \mathbf{H}^{-1}\mathbf{e}},$$

where:

$$H_{ij} = \mathrm{D}_{\mathrm{TV}}(D_i, D^\star)^2 + \mathrm{D}_{\mathrm{TV}}(D_j, D^\star)^2 - \mathrm{D}_{\mathrm{TV}}(D_i, D_j)^2,$$

and $\mathbf{e}$ is an $n$-dimensional vector with all entries equal to 1.

In the following steps, we provide a detailed proof of Theorem 3.

*Proof.* We begin by considering a binary classification problem using a linear model. For simplicity, assume the class labels $y \in \{-1, 1\}$ and input feature vectors $\boldsymbol{x} \in \mathbb{R}^d$. The objective is to predict the class label based on the input features.

In this proof, we employ Linear Discriminant Analysis (LDA). The goal of LDA is to find a linear decision boundary that separates the samples of different classes as effectively as possible. In LDA, the decision function is defined as:

$$f(\boldsymbol{x}) = \boldsymbol{\theta}^\top \boldsymbol{x} + b,$$

where the parameters $\boldsymbol{\theta}$ and $b$ are determined by:

$$\boldsymbol{\theta} = \boldsymbol{\Sigma}^{-1}(\boldsymbol{\mu}^{(1)} - \boldsymbol{\mu}^{(-1)}), \quad b = -\frac{1}{2}(\boldsymbol{\mu}^{(1)} + \boldsymbol{\mu}^{(-1)})^\top \boldsymbol{\Sigma}^{-1}(\boldsymbol{\mu}^{(1)} - \boldsymbol{\mu}^{(-1)}) + \ln\left(\frac{P(y=1)}{P(y=-1)}\right).$$

Here, $\boldsymbol{\mu}^{(1)}$ and $\boldsymbol{\mu}^{(-1)}$ are the mean vectors of the features for classes $y = 1$ and $y = -1$, respectively, and $\boldsymbol{\Sigma}$ is the shared covariance matrix of the features.

**Step 1: Assumptions**

**Assumption 7 (Class-Conditional Distribution).** *For each pre-trained dataset $D_i$ and the target dataset $D^\star$, the input features $\boldsymbol{x}$ are conditionally Gaussian given the class label $y$:*

*(a) For class $y = 1$:*

$$\boldsymbol{x} \mid y = 1 \sim \mathcal{N}(\boldsymbol{\mu}_i^{(1)}, \boldsymbol{\Sigma}), \quad \boldsymbol{x} \mid y = 1 \sim \mathcal{N}(\boldsymbol{\mu}^{\star(1)}, \boldsymbol{\Sigma}),$$

*where $\boldsymbol{\mu}_i^{(1)}$ and $\boldsymbol{\mu}^{\star(1)}$ are the mean vectors for class $y = 1$ in the pre-trained and target datasets, respectively.*

*(b) For class $y = -1$:*

$$\boldsymbol{x} \mid y = -1 \sim \mathcal{N}(\boldsymbol{\mu}_i^{(-1)}, \boldsymbol{\Sigma}), \quad \boldsymbol{x} \mid y = -1 \sim \mathcal{N}(\boldsymbol{\mu}^{\star(-1)}, \boldsymbol{\Sigma}),$$

*where $\boldsymbol{\mu}_i^{(-1)}$ and $\boldsymbol{\mu}^{\star(-1)}$ are the mean vectors for class $y = -1$.*

*(c) The covariance matrix $\boldsymbol{\Sigma}$ is shared across all datasets.*

**Assumption 8 (Covariance Matrix Properties).** *The class-conditional covariance matrix $\boldsymbol{\Sigma}$ satisfies:*

*(a) **Consistency:** The covariance matrices are the same for all pre-trained datasets $D_i$ and the target dataset $D^\star$:*
$$\boldsymbol{\Sigma}_i = \boldsymbol{\Sigma}^\star = \boldsymbol{\Sigma}.$$

*(b) **Positive Definiteness:** The covariance matrix $\boldsymbol{\Sigma}$ is positive definite:*
$$\boldsymbol{\Sigma} \succ 0,$$

*ensuring that $\boldsymbol{\Sigma}$ is invertible.*

**Step 2: Lemma in Linear Model**

**Lemma 2 (Parameter Distance and Total Variation).** *Under the above assumptions and given the linear model, the Euclidean distance between the parameters of the pre-trained models $\boldsymbol{\theta}_i$ and the target model $\boldsymbol{\theta}^\star$ is proportional to the total variation distance between their distributions:*

$$\|\boldsymbol{\theta}_i - \boldsymbol{\theta}^\star\| = C \cdot \mathrm{D}_{TV}(D_i, D^\star),$$

*where*

$$C = 2\sqrt{\frac{\tilde{\boldsymbol{\mu}}^\top \boldsymbol{\Lambda}^{-2} \tilde{\boldsymbol{\mu}}}{\tilde{\boldsymbol{\mu}}^\top \boldsymbol{\Lambda}^{-1} \tilde{\boldsymbol{\mu}}}} = 2\sqrt{\frac{\sum_{j=1}^d \frac{\tilde{\mu}_j^2}{\lambda_j^2}}{\sum_{j=1}^d \frac{\tilde{\mu}_j^2}{\lambda_j}}}$$

*is a constant, which depends on the eigenvalues of the covariance matrix $\boldsymbol{\Sigma}$ and the components of the transformed mean difference.*

*Proof of Lemma 2.* Given the LDA model, the parameters are related to the mean differences:

$$\boldsymbol{\theta}_i = \boldsymbol{\Sigma}^{-1}(\boldsymbol{\mu}_i^{(1)} - \boldsymbol{\mu}_i^{(-1)}), \quad \boldsymbol{\theta}^\star = \boldsymbol{\Sigma}^{-1}(\boldsymbol{\mu}^{\star(1)} - \boldsymbol{\mu}^{\star(-1)}).$$

The difference is:

$$\boldsymbol{\theta}_i - \boldsymbol{\theta}^\star = \boldsymbol{\Sigma}^{-1}(\boldsymbol{\mu}_i - \boldsymbol{\mu}^\star),$$

where $\boldsymbol{\mu}_i = \boldsymbol{\mu}_i^{(1)} - \boldsymbol{\mu}_i^{(-1)}$ and $\boldsymbol{\mu}^\star = \boldsymbol{\mu}^{\star(1)} - \boldsymbol{\mu}^{\star(-1)}$.

The Euclidean distance becomes:

$$\|\boldsymbol{\theta}_i - \boldsymbol{\theta}^\star\| = \sqrt{(\boldsymbol{\mu}_i - \boldsymbol{\mu}^\star)^\top \boldsymbol{\Sigma}^{-2} (\boldsymbol{\mu}_i - \boldsymbol{\mu}^\star)}.$$

For the total variation distance between the Gaussian distributions:

$$\mathrm{D}_{\mathrm{TV}}(D_i, D^\star) = \frac{1}{2}\|\boldsymbol{\mu}_i - \boldsymbol{\mu}^\star\|_{\boldsymbol{\Sigma}^{-1}} = \frac{1}{2}\sqrt{(\boldsymbol{\mu}_i - \boldsymbol{\mu}^\star)^\top \boldsymbol{\Sigma}^{-1} (\boldsymbol{\mu}_i - \boldsymbol{\mu}^\star)}.$$

Combining these, we derive the proportional relationship:

$$\|\boldsymbol{\theta}_i - \boldsymbol{\theta}^\star\| = C \cdot \mathrm{D_{TV}}(D_i, D^\star),$$

where

$$C = 2\sqrt{\frac{(\boldsymbol{\mu}_i - \boldsymbol{\mu}^\star)^\top \boldsymbol{\Sigma}^{-2}(\boldsymbol{\mu}_i - \boldsymbol{\mu}^\star)}{(\boldsymbol{\mu}_i - \boldsymbol{\mu}^\star)^\top \boldsymbol{\Sigma}^{-1}(\boldsymbol{\mu}_i - \boldsymbol{\mu}^\star)}}.$$

To express this constant $C$ further, we utilize the eigenvalue decomposition of the covariance matrix $\boldsymbol{\Sigma}$. Let:

$$\boldsymbol{\Sigma} = \mathbf{Q}\boldsymbol{\Lambda}\mathbf{Q}^\top,$$

where $\mathbf{Q}$ is the orthogonal matrix of eigenvectors of $\boldsymbol{\Sigma}$ and $\boldsymbol{\Lambda}$ is the diagonal matrix containing the eigenvalues $\lambda_1, \lambda_2, \ldots, \lambda_d$. Since $\boldsymbol{\Sigma}$ is positive definite, all eigenvalues $\lambda_j > 0$.

We rewrite the mean difference $\boldsymbol{\mu}_i - \boldsymbol{\mu}^\star$ in the eigenvector basis:

$$\tilde{\boldsymbol{\mu}} = \mathbf{Q}^\top(\boldsymbol{\mu}_i - \boldsymbol{\mu}^\star).$$

Here, $\tilde{\boldsymbol{\mu}}$ represents the coordinates of the mean difference in the space spanned by the eigenvectors of $\boldsymbol{\Sigma}$, and $\tilde{\mu}_j$ are its components.

The inverse and squared inverse of $\boldsymbol{\Sigma}$ are:

$$\boldsymbol{\Sigma}^{-1} = \mathbf{Q}\boldsymbol{\Lambda}^{-1}\mathbf{Q}^\top, \quad \boldsymbol{\Sigma}^{-2} = \mathbf{Q}\boldsymbol{\Lambda}^{-2}\mathbf{Q}^\top.$$

Substituting these into the expression for $C$, we get:

$$C = 2\sqrt{\frac{\tilde{\boldsymbol{\mu}}^\top \boldsymbol{\Lambda}^{-2} \tilde{\boldsymbol{\mu}}}{\tilde{\boldsymbol{\mu}}^\top \boldsymbol{\Lambda}^{-1} \tilde{\boldsymbol{\mu}}}}.$$

Writing out the components explicitly:

$$C = 2\sqrt{\frac{\sum_{j=1}^d \frac{\tilde{\mu}_j^2}{\lambda_j^2}}{\sum_{j=1}^d \frac{\tilde{\mu}_j^2}{\lambda_j}}}.$$

This final form shows that the proportionality constant $C$ depends on the eigenvalues of the covariance matrix $\boldsymbol{\Sigma}$ and the components $\tilde{\mu}_j$ of the transformed mean difference. It encapsulates how the mean differences project onto the eigenvectors of the covariance matrix. $\qquad\square$

**Step 3: Formulating the Quadratic Optimization Problem** Our objective is to determine the optimal combination coefficients $\gamma_i^\star$ that minimize the squared distance between the combined parameters $\boldsymbol{\theta}^{\mathrm{P}}$ and the target parameter $\boldsymbol{\theta}^\star$:

$$\min_\gamma \left\|\boldsymbol{\theta}^{\mathrm{P}} - \boldsymbol{\theta}^\star\right\|^2 = \min_\gamma \left\|\sum_{i=1}^n \gamma_i^\star \tilde{\boldsymbol{\theta}}_i - \boldsymbol{\theta}^\star\right\|^2,$$

subject to the constraints:

$$\sum_{i=1}^n \gamma_i^\star = 1, \quad \gamma_i^\star \geq 0.$$

Let $\boldsymbol{\delta}_i = \tilde{\boldsymbol{\theta}}_i - \boldsymbol{\theta}^\star$. Then, the objective function becomes:

$$\left\|\boldsymbol{\theta}^{\mathrm{P}} - \boldsymbol{\theta}^\star\right\|^2 = \left\|\sum_{i=1}^n \gamma_i^\star \boldsymbol{\delta}_i\right\|^2 = \sum_{i=1}^n \sum_{j=1}^n \gamma_i^\star \gamma_j^\star \boldsymbol{\delta}_i^\top \boldsymbol{\delta}_j.$$

Using the proportional relationship from Lemma 2, we express the inner product $\boldsymbol{\delta}_i^\top \boldsymbol{\delta}_j$ as:

$$\boldsymbol{\delta}_i^\top \boldsymbol{\delta}_j = \frac{1}{2}\left(\|\boldsymbol{\delta}_i\|^2 + \|\boldsymbol{\delta}_j\|^2 - \|\boldsymbol{\delta}_i - \boldsymbol{\delta}_j\|^2\right) = \frac{C^2}{2}\left(\mathrm{D_{TV}}(D_i, D^\star)^2 + \mathrm{D_{TV}}(D_j, D^\star)^2 - \mathrm{D_{TV}}(D_i, D_j)^2\right),$$

where $C$ is a positive constant of proportionality.

Define the symmetric matrix $\mathbf{H}$ with elements:

$$H_{ij} = \mathrm{D_{TV}}(D_i, D^\star)^2 + \mathrm{D_{TV}}(D_j, D^\star)^2 - \mathrm{D_{TV}}(D_i, D_j)^2.$$

Thus, the objective function simplifies to:

$$\left\| \boldsymbol{\theta}^{\mathrm{P}} - \boldsymbol{\theta}^\star \right\|^2 = \frac{C^2}{2} \boldsymbol{\gamma}^{*\top} \mathbf{H} \boldsymbol{\gamma}^\star.$$

Since $\frac{C^2}{2}$ is a positive constant, minimizing $\left\| \boldsymbol{\theta}^{\mathrm{P}} - \boldsymbol{\theta}^\star \right\|^2$ is equivalent to minimizing $\boldsymbol{\gamma}^{*\top} \mathbf{H} \boldsymbol{\gamma}^\star$. Therefore, the optimization problem becomes:

$$\min_{\boldsymbol{\gamma}^\star} \boldsymbol{\gamma}^{*\top} \mathbf{H} \boldsymbol{\gamma}^\star, \quad \text{subject to} \quad \sum_{i=1}^{n} \gamma_i^\star = 1, \quad \gamma_i^\star \geq 0.$$

**Step 4: Solving the Quadratic Optimization Problem**    We need to discuss by cases:

*Case 1: $n = 2$*

For $n = 2$, let $\gamma_2^\star = 1 - \gamma_1^\star$. Substituting into the objective function:

$$\boldsymbol{\gamma}^{*\top} \mathbf{H} \boldsymbol{\gamma}^\star = H_{11}(\gamma_1^\star)^2 + 2H_{12}\gamma_1^\star(1 - \gamma_1^\star) + H_{22}(1 - \gamma_1^\star)^2.$$

Expanding and simplifying:

$$\boldsymbol{\gamma}^{*\top} \mathbf{H} \boldsymbol{\gamma}^\star = (H_{11} + H_{22} - 2H_{12})(\gamma_1^\star)^2 + 2(H_{12} - H_{22})\gamma_1^\star + H_{22}.$$

To find the minimum, take the derivative with respect to $\gamma_1^\star$ and set it to zero:

$$\frac{d}{d\gamma_1^\star}(\boldsymbol{\gamma}^{*\top} \mathbf{H} \boldsymbol{\gamma}^\star) = 2(H_{11} + H_{22} - 2H_{12})\gamma_1^\star + 2(H_{12} - H_{22}) = 0.$$

Solving for $\gamma_1^\star$:

$$\gamma_1^\star = \frac{H_{22} - H_{12}}{H_{11} + H_{22} - 2H_{12}} = \frac{\mathrm{D_{TV}}(D_2, D^\star)^2 + \mathrm{D_{TV}}(D_1, D_2)^2 - \mathrm{D_{TV}}(D_1, D^\star)^2}{2\mathrm{D_{TV}}(D_1, D_2)^2}.$$

Thus, the optimal coefficients are:

$$\gamma_1^\star = \frac{\mathrm{D_{TV}}(D_2, D^\star)^2 + \mathrm{D_{TV}}(D_1, D_2)^2 - \mathrm{D_{TV}}(D_1, D^\star)^2}{2\mathrm{D_{TV}}(D_1, D_2)^2}, \quad \gamma_2^\star = 1 - \gamma_1^\star.$$

This solution is valid provided that $\gamma_1^\star, \gamma_2^\star \geq 0$.

*Case 2: $n > 2$*

When $n > 2$, an explicit solution for the optimal combination coefficients $\boldsymbol{\gamma}^\star$ generally does not exist under the constraints $\gamma_i^\star \geq 0$ due to the complexity introduced by multiple inequality constraints. However, if we further assume that all $\gamma_i^\star > 0$, we can derive an explicit solution.

Under the assumption $\gamma_i^\star > 0$ for all $i$, the optimization problem can be solved using the method of Lagrange multipliers. Construct the Lagrangian:

$$\mathcal{L}(\boldsymbol{\gamma}^\star, \lambda) = \boldsymbol{\gamma}^{*\top} \mathbf{H} \boldsymbol{\gamma}^\star - \lambda \left( \sum_{i=1}^{n} \gamma_i^\star - 1 \right).$$

Taking the derivative with respect to $\boldsymbol{\gamma}^\star$ and setting it to zero:

$$2\mathbf{H}\boldsymbol{\gamma}^\star - \lambda \mathbf{e} = 0 \quad \Rightarrow \quad \boldsymbol{\gamma}^\star = \frac{\lambda}{2} \mathbf{H}^{-1} \mathbf{e},$$

where $\mathbf{e}$ is an $n$-dimensional vector of ones.

Applying the constraint $\sum_{i=1}^{n} \gamma_i^\star = 1$:

$$\mathbf{e}^\top \gamma^\star = \frac{\lambda}{2} \mathbf{e}^\top \mathbf{H}^{-1} \mathbf{e} = 1 \quad \Rightarrow \quad \lambda = \frac{2}{\mathbf{e}^\top \mathbf{H}^{-1} \mathbf{e}}.$$

Substituting back, the optimal combination coefficients are:

$$\gamma^\star = \frac{\mathbf{H}^{-1} \mathbf{e}}{\mathbf{e}^\top \mathbf{H}^{-1} \mathbf{e}}.$$

This explicit solution holds provided that the matrix $\mathbf{H}$ is invertible and all resulting $\gamma_i^\star > 0$. If any $\gamma_i^\star \leq 0$, then numerical optimization methods must be employed to determine the optimal coefficients.

$\square$

# F  DETAILED EXPLANATION OF PARAMETER TRANSFORMATION

In this appendix, we provide a comprehensive theoretical exposition of the parameter transformation techniques introduced in Section 4.1.

## F.1  LEARNABLE WIDTH TRANSFORMATION

The width transformation is designed to adapt the weight matrices from a pre-trained source model to match the input and output dimensions of a target model, which may differ due to architectural changes. Given a weight matrix $\boldsymbol{\theta} \in \mathbb{R}^{d_{\text{in}} \times d_{\text{out}}}$ from a layer of the source model, our goal is to compute a transformed weight matrix $\tilde{\boldsymbol{\theta}} \in \mathbb{R}^{d'_{\text{in}} \times d'_{\text{out}}}$ suitable for the corresponding layer in the target model.

To facilitate this transformation, we introduce learnable transformation matrices $\mathbf{c}_{\text{in}} \in \mathbb{R}^{d'_{\text{in}} \times d_{\text{in}}}$ and $\mathbf{c}_{\text{out}} \in \mathbb{R}^{d'_{\text{out}} \times d_{\text{out}}}$. These matrices map the source input and output dimensions to the target dimensions, respectively. The transformed weight matrix is computed as:

$$\tilde{\boldsymbol{\theta}} = \mathbf{c}_{\text{in}} \boldsymbol{\theta} \mathbf{c}_{\text{out}}^\top.$$

The matrices $\mathbf{c}_{\text{in}}$ and $\mathbf{c}_{\text{out}}$ are treated as learnable parameters, optimized to minimize the loss function $\mathcal{L}$ of the target model. To provide a meaningful initialization that captures the most significant components of $\boldsymbol{\theta}$, we employ Singular Value Decomposition (SVD) on $\boldsymbol{\theta}$. Specifically, we decompose $\boldsymbol{\theta}$ as:

$$\boldsymbol{\theta} = \mathbf{U} \boldsymbol{\Sigma} \mathbf{V}^\top,$$

where: - $\mathbf{U} \in \mathbb{R}^{d_{\text{in}} \times r}$ contains the left singular vectors, - $\boldsymbol{\Sigma} \in \mathbb{R}^{r \times r}$ is a diagonal matrix of singular values, - $\mathbf{V} \in \mathbb{R}^{d_{\text{out}} \times r}$ contains the right singular vectors, - $r = \text{rank}(\boldsymbol{\theta})$.

To align the dimensions with the target model, we truncate or extend $\mathbf{U}$ and $\mathbf{V}$ to obtain $\tilde{\mathbf{U}} \in \mathbb{R}^{d_{\text{in}} \times r'}$ and $\tilde{\mathbf{V}} \in \mathbb{R}^{d_{\text{out}} \times r'}$, where $r' = \min(d'_{\text{in}}, d'_{\text{out}}, r)$. The truncated singular values are $\tilde{\boldsymbol{\Sigma}} \in \mathbb{R}^{r' \times r'}$. Formally:

$$\tilde{\mathbf{U}} = \mathbf{U}_{[:,1:r']}, \tilde{\boldsymbol{\Sigma}} = \boldsymbol{\Sigma}_{[1:r',1:r']}, \tilde{\mathbf{V}} = \mathbf{V}_{[:,1:r']}.$$

We initialize the transformation matrices $\mathbf{c}_{\text{in}}$ and $\mathbf{c}_{\text{out}}$ based on the truncated SVD components:

$$\mathbf{c}_{\text{in}}^{(0)} = \mathbf{W}_{\text{in}} \tilde{\mathbf{U}}^\top, \mathbf{c}_{\text{out}}^{(0)} = \mathbf{W}_{\text{out}} \tilde{\mathbf{V}}^\top,$$

where $\mathbf{W}_{\text{in}} \in \mathbb{R}^{d'_{\text{in}} \times r'}$ and $\mathbf{W}_{\text{out}} \in \mathbb{R}^{d'_{\text{out}} \times r'}$ are learnable weight matrices initialized randomly or based on heuristics.

Substituting and the transformed weight matrix becomes:

$$\tilde{\boldsymbol{\theta}} = \mathbf{W}_{\text{in}} \tilde{\mathbf{U}}^\top \boldsymbol{\theta} \tilde{\mathbf{V}} \mathbf{W}_{\text{out}}^\top.$$

Using the properties of SVD, we have:

$$\tilde{\mathbf{U}}^\top \boldsymbol{\theta} \tilde{\mathbf{V}} = \tilde{\mathbf{U}}^\top (\mathbf{U} \boldsymbol{\Sigma} \mathbf{V}^\top) \tilde{\mathbf{V}} = (\tilde{\mathbf{U}}^\top \mathbf{U}) \boldsymbol{\Sigma} (\mathbf{V}^\top \tilde{\mathbf{V}}) = \mathbf{I}_{r'} \tilde{\boldsymbol{\Sigma}} \mathbf{I}_{r'}^\top = \tilde{\boldsymbol{\Sigma}},$$

where $\mathbf{I}_{r'}$ is the identity matrix of size $r' \times r'$. Hence, the transformed weight matrix simplifies to:

$$\tilde{\boldsymbol{\theta}} = \mathbf{W}_{\text{in}} \tilde{\boldsymbol{\Sigma}} \mathbf{W}_{\text{out}}^\top.$$

This formulation decouples the adaptation process into learning $\mathbf{W}_{\text{in}}$ and $\mathbf{W}_{\text{out}}$, which project the truncated singular values to the target dimensions. Both $\mathbf{W}_{\text{in}}$ and $\mathbf{W}_{\text{out}}$ are learnable parameters optimized during training.

During training, the transformation matrices $\mathbf{c}_{\text{in}}$ and $\mathbf{c}_{\text{out}}$ are updated to minimize the loss function $\mathcal{L}$. The gradients with respect to these matrices are computed via backpropagation. For $\mathbf{c}_{\text{in}}$, the gradient is:

$$\frac{\partial \mathcal{L}}{\partial \mathbf{c}_{\text{in}}} = \frac{\partial \mathcal{L}}{\partial \tilde{\boldsymbol{\theta}}} \frac{\partial \tilde{\boldsymbol{\theta}}}{\partial \mathbf{c}_{\text{in}}},$$

where:

$$\frac{\partial \tilde{\boldsymbol{\theta}}}{\partial \mathbf{c}_{\text{in}}} = \boldsymbol{\theta} \mathbf{c}_{\text{out}}^\top.$$

Similarly, for $\mathbf{c}_{\text{out}}$:

$$\frac{\partial \mathcal{L}}{\partial \mathbf{c}_{\text{out}}} = \frac{\partial \mathcal{L}}{\partial \tilde{\boldsymbol{\theta}}} \frac{\partial \tilde{\boldsymbol{\theta}}}{\partial \mathbf{c}_{\text{out}}},$$

with:

$$\frac{\partial \tilde{\boldsymbol{\theta}}}{\partial \mathbf{c}_{\text{out}}} = (\mathbf{c}_{\text{in}} \boldsymbol{\theta})^\top.$$

Using these gradients, the transformation matrices are updated as:

$$\mathbf{c}_{\text{in}} \leftarrow \mathbf{c}_{\text{in}} - \eta \frac{\partial \mathcal{L}}{\partial \mathbf{c}_{\text{in}}}, \mathbf{c}_{\text{out}} \leftarrow \mathbf{c}_{\text{out}} - \eta \frac{\partial \mathcal{L}}{\partial \mathbf{c}_{\text{out}}},$$

where $\eta$ is the learning rate.

### F.2 LEARNABLE DEPTH TRANSFORMATION

The depth transformation adjusts the number of layers from $L$ in the source model to $L'$ in the target model. We introduce a learnable depth transformation matrix $\mathbf{D}_{\text{depth}} \in \mathbb{R}^{L' \times L}$, where each element $d_{ki}$ represents the learnable contribution of the $i$-th source layer to the $k$-th target layer.

The transformed parameters for the $k$-th target layer are computed as:

$$\tilde{\boldsymbol{\theta}}^k = \sum_{i=1}^{L} d_{ki} \boldsymbol{\theta}^i,$$

with the constraints:

$$d_{ki} \geq 0, \quad \sum_{i=1}^{L} d_{ki} = 1 \quad \forall k.$$

To satisfy the constraints, we parameterize $d_{ki}$ using the softmax function over learnable logits $\gamma_{ki}$:

$$d_{ki} = \frac{\exp(\gamma_{ki})}{\sum_{j=1}^{L} \exp(\gamma_{kj})}.$$

This formulation ensures that $d_{ki}$ are positive and sum to one for each $k$.

The logits $\gamma_{ki}$ are optimized alongside the model parameters by minimizing the overall loss $\mathcal{L}$. The gradient updates are:

$$\gamma_{ki} \leftarrow \gamma_{ki} - \eta \frac{\partial \mathcal{L}}{\partial \gamma_{ki}}.$$

The learnable coefficients $d_{ki}$ allow the model to dynamically determine the importance of each source layer for constructing the target layers.

## G ADDITIONAL RELATED WORK

Knowledge distillation (KD) (Hinton et al., 2015) is a widely used technique for transferring knowledge from a larger teacher model to a smaller student model by training the student to mimic the teacher's output logits or representations. The primary focus of KD is on transferring knowledge through the output space, aiming for model compression and efficiency without significant loss in performance.

Various extensions of KD have been proposed to improve efficiency and performance. Self-distillation (Zhang et al., 2019) involves training a model using its own outputs as soft targets, while mutual learning (Zhang et al., 2018) involves co-training multiple models to learn from each other. In the context of large-scale models, KD has been applied to compress transformer-based architectures (Sanh et al., 2019; Jiao et al., 2019) and to improve model generalization (Yuan et al., 2020).

Recent advances in KD have explored more sophisticated approaches. Cross-modal knowledge distillation (Gou et al., 2021) enables knowledge transfer between models operating on different modalities. Contrastive knowledge distillation (Tian et al., 2020) leverages contrastive learning to capture fine-grained structural knowledge. Additionally, adaptive knowledge distillation (Song et al., 2022) dynamically adjusts the distillation process based on the learning status of the student model.

While KD focuses on output-space knowledge transfer, our proposed method, SAIL, operates at the parameter level. SAIL directly transforms and integrates parameters from multiple pre-trained models to initialize a new model, leveraging the collective knowledge embedded in their parameters. This approach differs from KD in that it does not require training a student model to mimic a teacher's outputs; instead, it constructs a proximal parameter initialization that accelerates convergence during training.

Moreover, SAIL can be considered complementary to knowledge distillation. After applying SAIL to initialize the target model, KD can be employed as a subsequent optimization step to fine-tune or align the model to specific tasks. This combination could enhance both training efficiency and model performance by leveraging both parameter-space and output-space knowledge transfer.

## H EXPERIMENTAL DETAILS

### H.1 DATA DESCRIPTION

Our experiments primarily used the OpenWebText dataset, a large-scale corpus of web content. For cross-dataset generalization experiments, we also utilized the WikiText-103 dataset. Additionally, we conducted computer vision experiments using CIFAR-10, CIFAR-100, and Tiny ImageNet datasets.

**OpenWebText:** This dataset consists of web content extracted from URLs shared on Reddit. It contains a diverse range of topics and writing styles, making it suitable for training general-purpose language models. The dataset is stored in a binary format ('train.bin') where each token is represented as a 16-bit integer. Our preprocessed version of OpenWebText contains approximately 9 billion tokens.

**WikiText-103:** This dataset is derived from the set of verified Good and Featured articles on Wikipedia. It contains over 100 million tokens and serves as a high-quality benchmark for language modeling tasks. WikiText-103 is known for its long-term dependencies and diverse vocabulary, making it an excellent test for model generalization.

**Data Preprocessing for NLP Tasks:** For our experiments, we split the OpenWebText dataset into three subsets (D1, D2, and Dt) based on the mean token value of data blocks. This feature-based splitting approach ensures that each subset has a distinct distribution, allowing us to simulate different data domains. The splitting process is as follows:

1. We compute the mean token value for each block of 1024 tokens in the dataset.

2. We sort these blocks based on their mean token values.

3. We use the 33rd and 66th percentiles of these mean values as thresholds to split the data into three parts:
   - D1: Blocks with mean token values below the 33rd percentile
   - D2: Blocks with mean token values between the 33rd and 66th percentiles
   - Dt: Blocks with mean token values above the 66th percentile

This approach ensures that each subset has a distinct statistical distribution, simulating different data domains while still being part of the same overall corpus.

**T-SNE Visualizations:** To verify the effectiveness of our splitting approach and to visualize the distributions of different datasets, we performed more t-SNE (t-Distributed Stochastic Neighbor Embedding) analysis. Figure 4 presents a t-SNE visualization comparing samples from OpenWebText and WikiText-103, illustrating the distributional differences between these datasets.

**Computer Vision Datasets:** For our computer vision experiments, we used the following datasets:

- **CIFAR-10:** A dataset of 60,000 32x32 color images in 10 classes, with 6,000 images per class. There are 50,000 training images and 10,000 test images.

- **CIFAR-100:** Similar to CIFAR-10, but with 100 classes containing 600 images each. There are 500 training images and 100 testing images per class.

- **Tiny ImageNet:** A subset of ImageNet, consisting of 200 classes with 500 training images, 50 validation images, and 50 test images per class. Each image is 64x64 pixels.

These datasets were chosen to evaluate our method's performance across different levels of task complexity and dataset sizes in the computer vision domain.

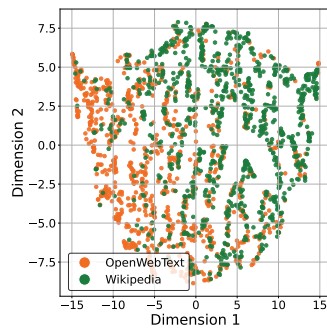

Figure 4: t-SNE visualization comparing OpenWebText and WikiText-103 samples

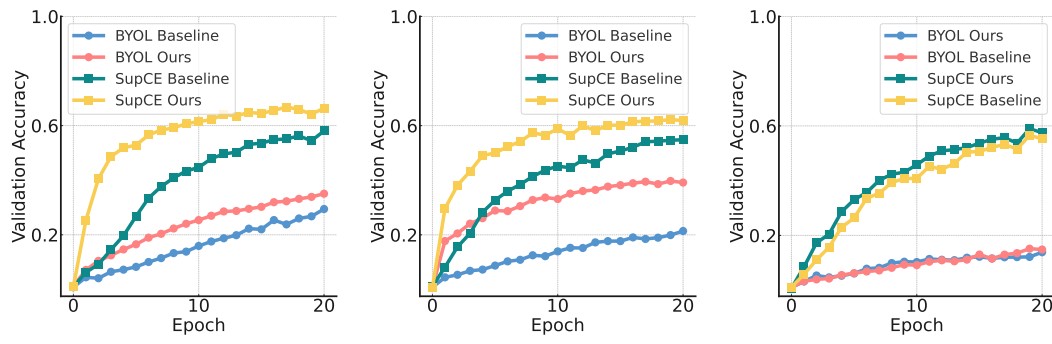

(a) ResNet-18 Modified (CIFAR-100)

(b) Standard ResNet-18 (CIFAR-100)

(c) ResNet-34 Modified (CIFAR-100)

Figure 5: **Accuracy in Different ResNet Configurations: (a)** Accuracy of ResNet-18 Modified trained with BYOL and SupCE on CIFAR-100. **(b)** Accuracy of standard ResNet-18 trained with BYOL and SupCE on CIFAR-100. **(c)** Accuracy of ResNet-34 Modified trained with BYOL and SupCE on CIFAR-100.

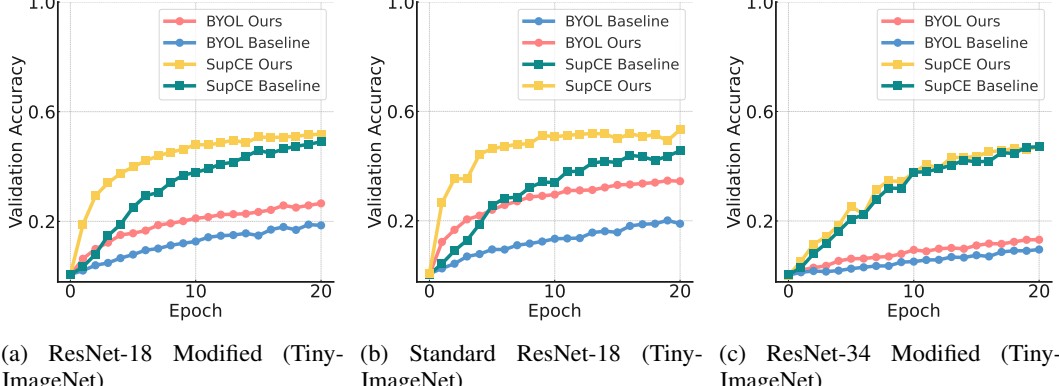

(a) ResNet-18 Modified (Tiny-ImageNet)

(b) Standard ResNet-18 (Tiny-ImageNet)

(c) ResNet-34 Modified (Tiny-ImageNet)

Figure 6: **Accuracy in Different ResNet Configurations: (a)** Accuracy of ResNet-18 Modified trained with BYOL and SupCE on Tiny-ImageNet. **(b)** Accuracy of standard ResNet-18 trained with BYOL and SupCE on Tiny-ImageNet. **(c)** Accuracy of ResNet-34 Modified trained with BYOL and SupCE on Tiny-ImageNet.

Detailed results for CIFAR-100 and Tiny ImageNet experiments are presented in Section H.2 of this appendix.

### H.2 Additional Experimental Results

## I Experiments in NLP

In this section, we present a comprehensive evaluation of our proposed method, **Sail**, in comparison with various baseline methods across multiple natural language processing (NLP) benchmarks. Leveraging the fully open **OLMo** framework, which includes model weights, training data, and evaluation tools, we ensure reproducibility and transparency in our experimental setup. We detail our experimental setup, including model configurations derived from the OLMo-1B and OLMo-7B variants, training procedures informed by our custom configuration file, hyperparameters, and dataset specifics. The results demonstrate the efficacy of **Sail** in enhancing model performance through optimal parameter merging and initialization.

### I.1 Experimental Setup

#### I.1.1 Models Used

We conducted our experiments using the following models from the OLMo Groeneveld et al. (2024):

**OLMo-1B**: A 1-billion parameter model pretrained on a diverse corpus, designed for general-purpose language understanding. **OLMo-7B**: A 7-billion parameter model with enhanced capabilities for complex language understanding and reasoning tasks. Each model variant is trained with distinct architectures, optimizers, and hardware configurations as specified in our training configuration file. The OLMo framework provides multiple checkpoints, enabling us to select intermediate states for parameter merging and initialization.

### I.1.2 CHECKPOINT SELECTION

For constructing the parameter set using **Sail**, we selected intermediate checkpoints based on the training progress captured in the OLMo:

- **OLMo-1B**: Intermediate checkpoints at steps 500,000 (`steps500000-2097B`), 600,000 (`steps600000-2517B`), and 700,000 (`steps700000-2936B`) were selected. These checkpoints represent different stages of model convergence and training dynamics.

- **OLMo-7B**: A single intermediate checkpoint at step 474,000 (`steps474000-2097B`) was selected, providing a reference point for evaluating larger model performance.

### I.1.3 TRAINING CONFIGURATION

Table 1 outlines the hyperparameters employed for training with **Sail**. These settings were chosen based on preliminary experiments and best practices in the literature to optimize model performance.

Table 1: Hyperparameters for **Sail**

| Hyperparameter | Value |
|---|---|
| Batch Size | 16 |
| Learning Rate | 4e-4 |
| Optimizer | AdamW |
| Number of Epochs | 1 |
| Weight Decay | 0.01 |
| Gradient Clipping | 1.0 |
| Scheduler | Cosine with Warmup |
| Warmup Steps | 2000 |

### I.2 DATASET DETAILS

We evaluated our models on a diverse set of NLP benchmarks to ensure a comprehensive assessment of **Sail**'s capabilities. The datasets encompass a range of tasks, including commonsense reasoning, question answering, and causal reasoning. Below are the details of each dataset used:

- **PIQA**:Bisk et al. (2020) Physical commonsense reasoning with 7,000 training examples and 1,500 test examples.

- **HellaSwag**:Zellers et al. (2019) Complex multiple-choice questions requiring robust inference, consisting of 70,000 training examples and 10,000 test examples.

- **Winogrande**:ai2 (2019) Pronoun resolution with 44,000 training examples and 8,000 test examples.

- **SciQ**:Johannes Welbl (2017) Comprehension of scientific texts, containing 13,679 training examples and 1,384 test examples.

- **ARC-Easy**:Clark et al. (2018) Grade-school level science questions with 3,779 training examples and 1,366 test examples.

- **COPA**:Roemmele et al. (2011) Causal reasoning by selecting plausible alternatives, comprising 1,000 training examples and 500 test examples.

### I.3 COMPLETE RESULTS

We present a comprehensive comparison of **Sail** against various baseline methods across all evaluated NLP benchmarks. The results are consolidated in Table 2, demonstrating the superior performance and flexibility of **Sail** in model initialization and parameter merging.

Table 2: Comparison of **Sail** with Baseline Methods (Accuracy %)

| Dataset | Train from Scratch | LIGO | Uniform Soup | Greedy Soup | Sail (Ours) |
|---|---|---|---|---|---|
| PIQA | 51.96 | 52.29 | 54.80 | 57.73 | **61.92** |
| HellaSwag | 24.87 | 25.33 | 25.25 | 27.65 | **34.48** |
| Winogrande | 51.14 | 50.20 | 50.51 | 52.09 | **52.96** |
| SciQ | 22.10 | 23.90 | 51.90 | 59.70 | **70.30** |
| ARC-Easy | 27.54 | 29.65 | 27.19 | 34.91 | **42.63** |
| COPA | 58.00 | 55.00 | 57.00 | 51.00 | **63.00** |

**Comparison of Sail with Baseline Methods** These curves display perplexity across step for models initialized with **Sail** compared to those with random initialization. The plots confirm that **Sail** not only achieves higher final performance but also converges more rapidly during training, consistent with our findings in computer vision (CV) experiments.

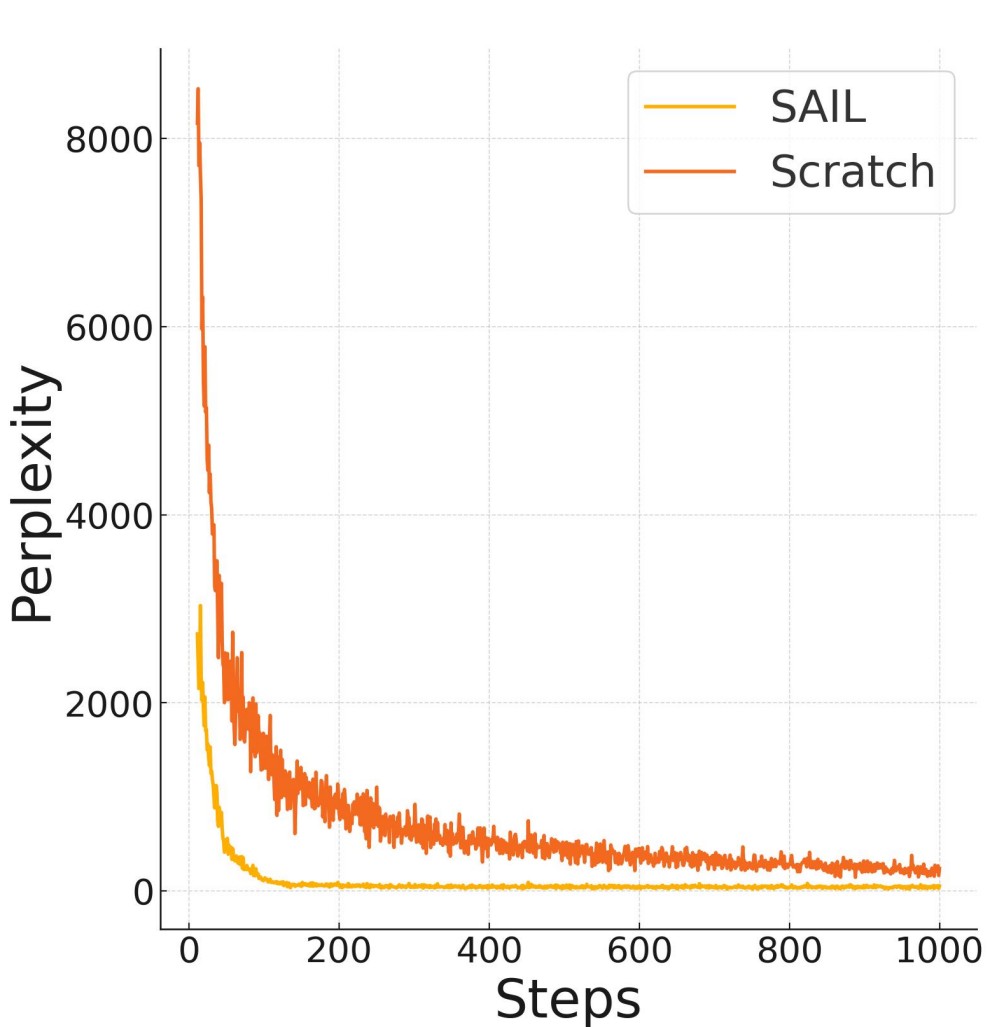

Figure 7: Perplexity for Models Initialized with **Sail** vs. Random Initialization on NLP Benchmarks.

## J  SAIL FOR SiT DIFFUSION MODELS

### J.1  APPLYING SAIL TO SiT DIFFUSION MODELS

In this section, we extend our Structured-Initialization Learning (SAIL) framework to accelerate the training of state-of-the-art SiT (Scalable Interpolant Transformers) diffusion models (Ma et al., 2024), which are generative models. By adapting SAIL to SiT diffusion models, we aim to demonstrate the versatility of our method in different domains and its effectiveness in improving training efficiency.

In visual representation learning, aligning the representations within generative models with pre-trained ones improves both semantic integration and performance (Yu et al., 2024). Within our SAIL framework, this alignment is achieved through specific adaptations. One such adaptation is Latent-to-Representation alignment, which serves as a case study for applying SAIL to SiT models, given that SiT training occurs in latent space of VAE (Ma et al., 2024).

Formally, let us denote:

- $\mathcal{Z}$ as the latent space of the diffusion model.
- $\mathcal{R}$ as the external pre-trained representation space.
- $f_{\mathrm{P}} : \mathcal{Z} \to \mathcal{R}$ as the pre-trained representation model.
- $f_{\mathrm{A}} : \mathcal{H} \to \mathcal{R}$ as the alignment function within SAIL, where $\mathcal{H}$ represents the hidden representations of the diffusion model.

The objective is to minimize the discrepancy between the representations derived from the VAE latent space and those from the pre-trained representation space, ensuring coherent semantic alignment within the SAIL framework.

### J.2  WEIGHT INITIALIZATION IN SAIL FOR SiT MODELS

Using SAIL, we initialize the weights of the SiT diffusion transformer by leveraging pre-trained models. This corresponds to our parameter transformation technique, where we adjust the dimensions of pre-trained model parameters to match the target SiT architecture.

Formally, let $\boldsymbol{\theta}_{\mathrm{SAIL}}^{\mathrm{P}}$ represent the pre-trained weights obtained by optimizing the alignment between latent variables and pre-trained representations:

$$\boldsymbol{\theta}_{\mathrm{SAIL}}^{\mathrm{P}} = \arg\min_{\boldsymbol{\theta}} \mathcal{L}_{\mathrm{Align}}(\boldsymbol{\theta}), \tag{14}$$

where $\mathcal{L}_{\mathrm{Align}}$ is the alignment loss function within the SAIL framework that measures the discrepancy between the model's latent representations and the pre-trained representation space. By initializing the SiT model with $\boldsymbol{\theta}_{\mathrm{SAIL}}^{\mathrm{P}}$, we ensure that the model starts with parameters that already encode meaningful semantic information, thereby enhancing the efficiency of subsequent training stages.

### J.3  INCORPORATING ALIGNMENT LOSS IN SAIL TRAINING

In addition to weight initialization, we incorporate an alignment loss term into the SAIL training objective to continuously align the model's hidden representations with the pre-trained representations. This strategy complements our proximal parameter integration and retraining approach, which efficiently combines transformed parameters to initialize new models.

The total loss function during training becomes:

$$\mathcal{L}_{\mathrm{Total}} = \mathcal{L}_{\mathrm{Velocity}} + \lambda_{\mathrm{REPA}} \mathcal{L}_{\mathrm{REPA}} + \lambda_{\mathrm{Align}} \mathcal{L}_{\mathrm{Align}}, \tag{15}$$

where:

- $\mathcal{L}_{\mathrm{Velocity}}$ is the primary loss for velocity prediction in the diffusion model.
- $\mathcal{L}_{\mathrm{REPA}}$ is the representation alignment loss as defined in REPA (Yu et al., 2024).

- $\mathcal{L}_{\text{Align}}$ is the alignment loss within SAIL.
- $\lambda_{\text{REPA}}$ and $\lambda_{\text{Align}}$ are hyperparameters controlling the strength of each alignment component.

The alignment loss is defined as:

$$\mathcal{L}_{\text{Align}} = \mathbb{E}_{\mathbf{z}_t, \mathbf{h}_t} \left[ \| f_{\text{P}}(\mathbf{z}_t) - f_{\text{A}}(\mathbf{h}_t) \|^2 \right], \tag{16}$$

where:

- $\mathbf{z}_t$ represents the latent variables at time $t$.
- $\mathbf{h}_t$ represents the hidden states of the model at time $t$.

### J.4 EXPERIMENTAL SETUP

To evaluate the effectiveness of integrating Latent-to-Representation (L2R) alignment within the SAIL framework for improving SiT pre-training, we conduct a series of experiments on the ImageNet $256 \times 256$ dataset. Our primary objective is to assess how the incorporation of L2R influences both the training efficiency and the quality of the generated representations.

We utilize the ImageNet dataset, specifically the $256 \times 256$ resolution subset, which contains 1.28 million training images and 50,000 validation images across 1,000 classes. All images are resized to $256 \times 256$ pixels and normalized using standard ImageNet statistics. Data augmentation techniques, including random horizontal flipping and random cropping, are employed to enhance the diversity of the training data.

The SiT-B/2 model, as described by Ma et al. (2024), serves as our baseline architecture. We enhance the training of this model by integrating our SAIL outlined in the previous sections. Specifically, the L2R model was initialized with pre-trained weights obtained from an alignment task between latent variables and pre-trained representations, ensuring that the initial parameters encoded meaningful semantic information.

#### J.4.1 HYPERPARAMETERS

Following previous studies (Ma et al., 2024; Yu et al., 2024), the key hyperparameters for our experiments are summarized in Table 3.

Table 3: Hyperparameters used for training SAIL with L2R model on ImageNet $256 \times 256$.

| Hyperparameter | Value |
|---|---|
| Learning Rate | $1 \times 10^{-4}$ |
| Optimizer | AdamW |
| $\beta_1$ | 0.9 |
| $\beta_2$ | 0.999 |
| Weight Decay | 0.01 |
| Batch Size | 256 |
| Training Iterations | 400K |
| Gradient Clipping Norm | 1.0 |
| $\lambda_{\text{REPA}}$ | 0.5 |
| $\lambda_{\text{L2R}}$ | 0.5 |
| Latent Scale | 0.18215 |
| Latent Bias | 0.0 |

### J.5 RESULTS

The integration of our SAIL framework significantly improve both the training efficiency and effectiveness of SiT. Table 4 provides a comparative analysis between the baseline SiT model and the enhanced version incorporating REPA and SAIL across various training iterations.

Table 4: Performance comparison between the baseline SiT model and the SiT model enhanced with REPA and SAIL at various training iterations over ImageNet $256 \times 256$ generation. Improvements with SAIL over REPA are indicated with arrows and highlighted in red.

| Model | #Params | Iter. | FID↓ | sFID↓ | IS↑ | Prec.↑ |
|---|---|---|---|---|---|---|
| SiT-B/2 (Ma et al., 2024) | 130M | 400K | 33.0 | 6.46 | 43.7 | 0.53 |
| + REPA | 130M | 50K | 78.2 | 11.71 | 17.1 | 0.33 |
| + SAIL (ours) | 130M | 50K | 67.6 (↓10.6) | 16.19 (↑4.48) | 20.5 (↑3.4) | 0.34 (↑0.01) |
| + REPA | 130M | 100K | 49.5 | 7.00 | 27.5 | 0.46 |
| + SAIL (ours) | 130M | 100K | 35.9 (↓13.6) | 7.02 (↓0.02) | 45.1 (↑17.6) | 0.53 (↑0.07) |
| + REPA | 130M | 200K | 33.2 | 6.68 | 43.7 | 0.54 |
| + SAIL (ours) | 130M | 200K | 19.8 (↓13.4) | 6.15 (↓0.53) | 81.9 (↑38.2) | 0.64 (↑0.10) |
| + REPA | 130M | 400K | 24.4 | 6.40 | 59.9 | 0.59 |
| + SAIL (ours) | 130M | 400K | 12.2 (↓12.2) | 5.90 (↓0.50) | 119.4 (↑59.5) | 0.70 (↑0.11) |

## K    CONTROL EXPERIMENTS WITH THE SPIRALS DATASET

To empirically validate our theoretical findings and demonstrate the practical effectiveness of the SAIL method, we conduct control experiments using the Spirals dataset. By considering different model initializations and non-overlapping data distributions, we aim to verify that our theoretical predictions hold in practice when applied to multi-layer perceptrons (MLPs).

The Spirals dataset is a synthetic dataset where data points are arranged in two interleaving spirals, forming a challenging classification problem that requires models to learn complex, non-linear decision boundaries (Guyon & Elisseeff, 2003). This dataset is well-suited for assessing the capability of models to capture intricate patterns and for evaluating the effectiveness of initialization strategies in non-convex optimization landscapes.

We design our experiments to achieve two main objectives:

1. **Faster Optimization Speed**: Demonstrate that models initialized with the SAIL method converge faster than those with random initialization.
2. **Effectiveness of SAIL under Different Data Distributions**: Assess the impact of using pre-trained models trained on different, non-overlapping subsets of the Spirals dataset to evaluate the limitations of the SAIL method.

### K.1    EXPERIMENTAL SETUP

We generate the Spirals dataset $\mathcal{D}^\star$ consisting of data points from two interleaving spirals, with each spiral representing a distinct class. To create different, non-overlapping data distributions, we derive two additional separate subsets from $\mathcal{D}^\star$:

- $\mathcal{D}_1$: Contains data points exclusively from the first spiral, comprising the first 40% of the training data.
- $\mathcal{D}_2$: Contains data points exclusively from the second spiral, comprising the next 40% of the training data.

We train two models separately on these non-overlapping subsets:

- $\theta_1$: Trained on $\mathcal{D}_1$.
- $\theta_2$: Trained on $\mathcal{D}_2$.

Using the SAIL method, we transform and merge the parameters of $\theta_1$ and $\theta_2$ to form the proximal parameter $\theta^P$, which serves as the initialization for training the target model on the full dataset $\mathcal{D}^\star$.

We compare the performance of models initialized with $\theta^P$ against models with random initialization and models trained on $\mathcal{D}^\star$ from scratch. All models are trained using the same neural network architecture: a multi-layer perceptron (MLP) with one hidden layer of 50 neurons and ReLU activation functions. The learning rate is set to $5 \times 10^{-3}$, and models are trained using the Adam optimizer for 300 epochs.

We evaluate the models based on three metrics: training loss, accuracy, and gradient norm. The gradient norm provides insight into the stability and efficiency of the optimization process.

**Faster Optimization Speed**    Figure 8 shows the training loss and accuracy over epochs for the models initialized with $\theta^P$ and with random initialization. The model initialized with $\theta^P$ converges to a lower loss significantly faster than the randomly initialized model. The accuracy of the model with SAIL initialization improves rapidly and reaches a higher final accuracy compared to the model with random initialization. This demonstrates that the SAIL initialization provides a better starting point in the parameter space, closer to the optimum, thus requiring fewer iterations to converge.

**Gradient Norm Analysis**    Figure 9 presents the gradient norm over epochs. The SAIL-initialized model exhibits a smaller gradient norm earlier in training, indicating a more stable optimization process. This suggests that the model starts closer to a region with flatter loss landscape, facilitating more efficient convergence compared to the randomly initialized model.

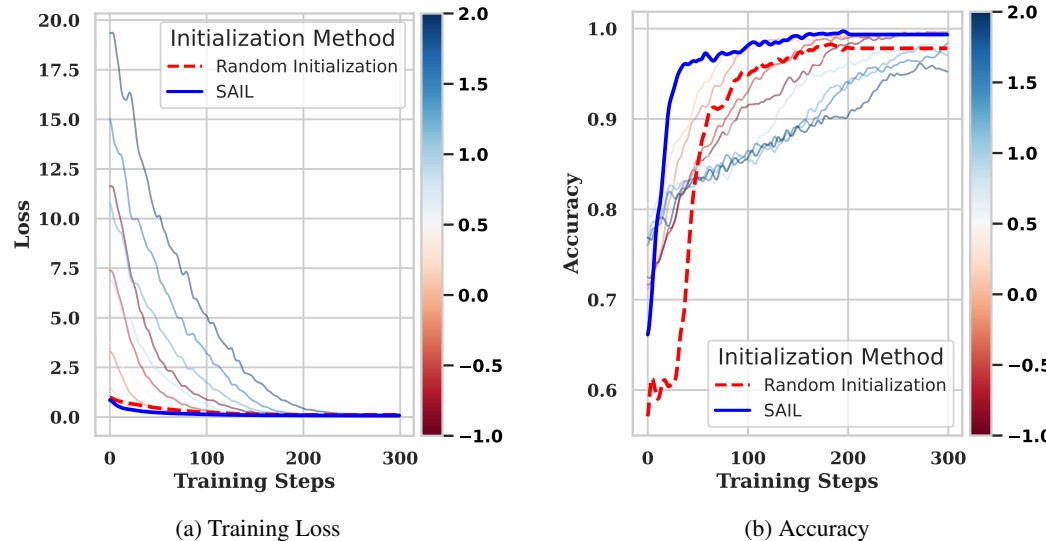

(a) Training Loss                    (b) Accuracy

Figure 8: Comparison of training loss and accuracy over epochs for models initialized with $\theta^{\mathrm{P}}$ (SAIL) and with random initialization. The SAIL-initialized model converges faster and achieves better performance. The color bar indicates the value of $\gamma$.

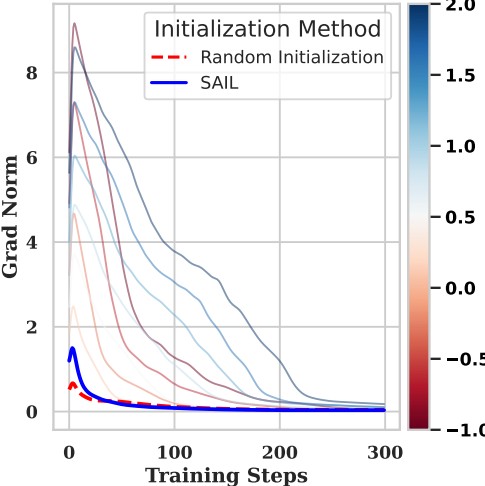

Figure 9: Gradient norm of the parameters over epochs for models initialized with $\theta^{\mathrm{P}}$ (SAIL) and with random initialization. The SAIL-initialized model demonstrates a more stable optimization trajectory. The color bar indicates the value of $\gamma$.

**Effectiveness of SAIL under Different Data Distributions**   To assess the robustness of the SAIL method, we conducted experiments where the pre-trained models $\theta_1$ and $\theta_2$ were trained on different, non-overlapping subsets of $\mathcal{D}_1$ and $\mathcal{D}_2$. In this scenario, the benefits of the SAIL initialization are influenced by the disparity between the pre-trained models' data distributions and that of the target dataset $\mathcal{D}^\star$.

As shown in   Figure 8   and   Figure 9 , even when the pre-trained models are trained on distinct and non-overlapping subsets, the SAIL initialization still provides a convergence speed advantage compared to random initialization, albeit reduced compared to the scenario where the pre-trained models are trained on larger or more representative portions of the target dataset.

These experiments confirm that the SAIL method effectively accelerates the convergence of MLP models on complex, non-convex tasks like the Spirals dataset. By initializing the model parameters

with the proximal parameter $\theta^{\mathrm{P}}$ derived from pre-trained models on related data, we achieve faster optimization and better final performance compared to random initialization. The method remains effective even when the pre-trained models are trained on different, non-overlapping data distributions, demonstrating the versatility and robustness of SAIL.

