# OpenReview forum: "Structured-Initialization Learning"
_ICLR.cc/2025/Conference — Submitted to ICLR 2025_

### Official Review · Reviewer_WX7E · 2024-10-31

**Soundness:** 3
**Presentation:** 3
**Contribution:** 3
**Rating:** 8
**Confidence:** 4

**Summary:**

This paper proposed a novel way to combine the weights of multiple pre-trained LLMs into a smaller model. The problem to be addressed is important, as it offers opportunity to train a smaller specialized model faster, rather than train from scratch.

The paper proposed a set of theorems support it's claims, which offers theoretical guarantees.

**Strengths:**

- This paper supports its claims with both theoretical and empirical results. The proposed theorems are proved in the appendix, which looks correct (without in-depth review)

- This paper address an important problem of LLM reusing, which lacks efficient solutions before. The proposed method seems to be better than naive averaging weights.

- This paper marks different theorems and labels in color, which helps readers to quickly locate and navigate.

**Weaknesses:**

- The field of knowledge distillation should be discussed in the related works section, which is highly related.

- The use of colored block seems to be too extensive, which is not common in research papers. Though not an outstanding weakness point.

**Questions:**

- Why is there a spike at $\gamma=0.5$?

- Are you assuming all other network elements are the same, e.g. activation functions, vocabulary size?

- Better to add discussion about knowledge distillation.

---

> ### Author Response · Authors · 2024-11-22
>
> ### **Weaknesses:**
>
> > W1: **The field of knowledge distillation should be discussed in the related works section, which is highly related & Better to add discussion about knowledge distillation.**
> >
>
> **Our Response:**
>
> We appreciate the reviewer highlighting the relevance of knowledge distillation to our work. In response, we have expanded the related work in **Appendix G** **in page of 32** to include a detailed discussion on knowledge distillation.
>
> **Comparison Between Knowledge Distillation and SAIL:**
>
> - **Knowledge Distillation:** This technique involves training a smaller student model to replicate the behavior of a larger teacher model by mimicking its output logits. The primary focus is on transferring output-space knowledge to achieve model compression and efficiency.
> - **SAIL (Our Approach):** Unlike knowledge distillation, SAIL focuses on parameter-space knowledge transfer. It directly transforms and integrates parameters from multiple pre-trained models to initialize a new model. This method leverages the collective knowledge embedded in the parameters, rather than relying solely on output alignment.
>
> **Complementary Nature:**
>
> While both approaches aim to transfer knowledge from existing models, they operate at different levels. Moreover, knowledge distillation can be utilized as a subsequent optimization step following SAIL, where the merged parameters are further fine-tuned or re-trained to align with the target task's specific requirements.
>
> > W2: The use of colored block seems to be too extensive, which is not common in research papers. Though not an outstanding weakness point.
> >
>
> **Our Response:**
>
> We appreciate your comment regarding the colored blocks. While we aimed to enhance the readability and highlight key conclusions for readers, we acknowledge your point about academic convention. We will consider adopting a more standard formatting approach further.
>
> ### **Questions:**
>
> > Q1: **Why is there a spike at γ=0.5?**
> >
>
> **Our Response:**
>
> We acknowledge the reviewer's observation regarding the spike at $\gamma=0.5$ in the validation loss curves depicted in **Figure 2b**. This spike is indicative of the complex dynamics involved in merging model parameters and can be attributed to the following factors:
>
> ### **Explanation for the Spike at $\gamma=0.5$:**
>
> 1. **Interference Between Divergent Representations:**
>     - At $\gamma=0.5$, the parameters from the two pre-trained models are weighted equally. If the models have learned significantly different or even conflicting representations due to their training on distinct datasets (as is the case with $D_1$ and $D_2$), the direct average can result in parameter interference.
>     - This interference can temporarily degrade the model's performance, manifesting as an increase in validation loss.
> 2. **Lack of Alignment in Parameter Space:**
>     - Equal weighting does not account for the relative relevance or compatibility of each model's parameters with respect to the target dataset $D_t$.
>     - Without proper alignment, the superposition of parameters may not produce a coherent initial model, leading to suboptimal performance at $\gamma=0.5$.
>
>
> > Q2: **Are you assuming all other network elements are the same, e.g., activation functions, vocabulary size?**
> >
>
> **Our Response:**
>
> Yes, in our experiments, there is usually a target model architecture and a corresponding activation function or vocabulary size, and we directly choose to find the vocabulary size of a target model size in our experiments and directly inherit it, without converting it here.

---

> > ### Author Response · Authors · 2024-12-02
> >
> > Dear Reviewer,
> >
> > As the deadline approaches, we kindly request your response to our feedback at your earliest convenience. Thank you for your time and consideration.
> >
> > Best regards

---

> > > ### Author Response · Authors · 2024-12-03
> > >
> > > Dear Reviewer,
> > >
> > > We sincerely appreciate your detailed review of our manuscript. Given the limited time remaining in the discussion phase, we would welcome the opportunity to promptly address your concerns and discuss any additional questions you may have to ensure our responses are thorough and satisfactory.
> > >
> > > We look forward to your timely feedback.
> > >
> > > Best regards,
> > > Authors

---

### Official Review · Reviewer_1vMa · 2024-11-01

**Soundness:** 2
**Presentation:** 3
**Contribution:** 2
**Rating:** 6
**Confidence:** 3

**Summary:**

This paper proposes Structured-Initialization Learning (SAIL), a method to accelerate training for large models by reusing parameters from pre-trained models. The approach includes transforming parameters to fit the target model and integrating them to form a more optimal starting point for training, reducing the need for random initialization.

**Strengths:**

1. The paper provides a solid theoretical analysis, effectively demonstrating how the proposed Proximal Parameter initialization leads to faster convergence. The authors present well-structured convergence theorems, lending strong support to the efficacy of SAIL in reducing training time and improving efficiency.
2. The method is tested on both NLP and computer vision tasks, demonstrating applicability across different domains and model architectures.

**Weaknesses:**

1. **Limited Novelty in Leveraging Pre-trained Models for Initialization** The proposed method of reusing parameters from pre-trained models to accelerate the training of new models is similar to existing work [1-2].
2. The motivation and system design of Figure 1 claims to use a pre-trained model such as LLM, however, the actual experiments are conducted by training the model from scratch on small-scale datasets in a controlled setup. It would be good to actually use the pre-trained models in hugging faces to create the SAIL.
3. The explanation of how the parameter transformation is conducted is not clear enough, the authors mentioned the random projection and learnable methods, but a detailed investigation and experiments on which method is better are not included.


**Reference**

[1] Initializing Models with Larger Ones ICLR 2024
[2] Scaling Smart: Accelerating Large Language Model Pre-training with Small Model Initialization

**Questions:**

1. Could the author provide the training curves of the NLP experiments as shown in the CV experiments, it helps to verify if the proved fast convergence is also applicable to transformer training.

---

> ### Author Response · Authors · 2024-11-22
> **Official Comment by Authors (1/2)**
>
> ### Weaknesses:
>
> > W1: **Limited novelty in leveraging pre-trained models for initialization; similarity to existing work [1][2].**
> >
>
> **Our Response:**
>
> We appreciate the reviewer’s feedback regarding the novelty of our approach in leveraging pre-trained models for initialization and the perceived similarity to existing works [1][2]. We would like to clarify the distinct contributions and theoretical advancements that **Structured Initialization Learning (SAIL)** introduces, setting it apart from prior methods. **And we will expand our discussion with [1][2] in our manuscript. We have noticed that [2] is concurrent work with ours and hasn't been open-sourced.**
>
> ### **Distinction from Existing Work:**
>
> 1. **Comprehensive Parameter Transformation and Architecture Alignment:**
>     - **SAIL** introduces a **dual-faceted parameter transformation technique** that adjusts both the width (number of neurons per layer) and depth (number of layers) of pre-trained models to match the architecture of the target model. This allows for the integration of multiple pre-trained models with **varying architectures and sizes**, enabling a seamless amalgamation of diverse knowledge bases.
>     - *In contrast*, **Xu et al. (2023)** [1] propose a method called **Weight Selection**, which primarily focuses on initializing smaller models by selecting subsets of weights from a larger pre-trained model. Their approach is limited to models within the same family and requires similar architectures, emphasizing downscaling rather than flexible architectural alignment.
>     - **Samragh et al. (2024)** [2] introduce **HyperCloning**, a method that expands a small pre-trained model to a larger one through weight replication and symmetric initialization. Their technique involves duplicating neurons to match the larger model's dimensions but does not account for architectural differences beyond scaling width dimensions.
>     - **In SAIL**, we formulate the parameter transformation as a linear mapping  $T_i$  for each pre-trained model parameter  $\theta_i$ :
>
>          $\tilde{\theta}_i = T_i(\theta_i) \in \mathbb{R}^d$
>
>         where  $d$  is the dimensionality of the target model's parameter space.
>
>     - This transformation ensures that the parameters from different pre-trained models are projected into a common space, allowing for seamless integration regardless of their original architectures.
> 2. **Optimal Parameter Integration with Theoretical Guarantees:**
>     - **SAIL** defines the **Proximal Parameter**  $\theta^P$  as an optimal linear combination of transformed parameters:
>
>          $\theta^P = \sum_{i=1}^K \gamma_i^* \tilde{\theta}_i$
>
>         where $\gamma_i^*$ are the combination coefficients optimized to minimize the distance between  $\theta^P$  and the target model's optimal parameters  $\theta^*$ .
>
>     - **Our Theorem 3** provides explicit solutions for  $\gamma_i^*$  by solving:
>
>          $\gamma^* = \arg\min_{\gamma} \left\| \sum_{i=1}^K \gamma_i \tilde{\theta}_i - \theta^* \right\|^2$
>
>         subject to  $\sum_{i=1}^K \gamma_i = 1$  and  $\gamma_i \geq 0$ .
>
>     - The coefficients  $\gamma_i^*$  are determined based on the statistical distances (e.g., total variation distance) between the distributions of the pre-trained models and the target data, providing a theoretically optimal integration of knowledge.
>     - *In contrast*, **Xu et al. (2023)** and **Samragh et al. (2024)** do not formulate an optimization problem for parameter integration. Their methods lack theoretical guarantees regarding the optimality of parameter initialization and convergence speed.
> 3. **Theoretical Convergence Analysis:**
>     - **SAIL** offers rigorous theoretical analysis demonstrating that initializing with the Proximal Parameter  $\theta_P$  leads to faster convergence to the optimal parameters  $\theta^*$  compared to random initialization.
>     - **Theorem 2** in our work shows that the suboptimality after  $T$  iterations satisfies:
>
>         $J(\theta^{(T)}) - J(\theta^*) \leq (1 - \eta \mu)^T \left( J(\theta^P) - J(\theta^*) \right)$
>
>         where  $\eta$  is the learning rate and  $\mu$  is the strong convexity parameter of the loss function  $J(\theta)$ .
>
>     - This result indicates that by minimizing the initial suboptimality  $J(\theta^P) - J(\theta^*)$ through optimal parameter integration, **SAIL** accelerates the convergence of the training process.
>     - *Existing works* do not provide such convergence analysis or theoretical foundations for their initialization methods.

---

> ### Author Response · Authors · 2024-11-22
> **Official Comment by Authors (2/2)**
>
> 4. **Integration of Multiple Pre-trained Models Beyond Single-Model Scaling:**
>     - **SAIL** is designed to integrate knowledge from multiple pre-trained models, potentially trained on different datasets or tasks, to form a more comprehensive initialization.
>     - **Xu et al. (2023)** focus on transferring weights from a single larger model to a smaller one via weight selection. Their method does not generalize to integrating multiple models.
>     - **Samragh et al. (2024)** scale up a single small model through HyperCloning, which involves duplicating weights symmetrically but does not consider merging different models.
>
> **References:**
>
> [1] Xu Z, Chen Y, Vishniakov K, et al. Initializing models with larger ones[C]//The Twelfth International Conference on Learning Representations. 2023.
>
> [2] Samragh M, Mirzadeh I, Vahid K A, et al. Scaling smart: Accelerating large language model pre-training with small model initialization[J]. arXiv preprint arXiv:2409.12903, 2024.
>
> > W2: **The motivation and system design of Figure 1 claims to use a pre-trained model such as LLM, but experiments are conducted by training the model from scratch on small-scale datasets in a controlled setup. It would be good to actually use the pre-trained models in Hugging Face to create the SAIL.**
> >
>
> **Our Response:**
>
> We appreciate the reviewer’s suggestion. In our initial submission, we focused on controlled experiments to isolate and understand the effects of **SAIL**. To enhance the robustness of our work, we have now incorporated experiments using pre-trained models from Hugging Face, specifically the **OLMo** model. The detailed results are presented in **Appendix I in page of 34**.
>
> > W3: **The explanation of how the parameter transformation is conducted is not clear enough; the authors mentioned random projection and learnable methods, but a detailed investigation and experiments on which method is better are not included.**
> >
>
> **Our Response:**
>
> Thank you for highlighting the need for a clearer explanation of our parameter transformation process.
>
> In our work, we utilize a **learnable linear projection** method to map weights from pre-trained models to the target model architecture. We have updated the manuscript to include a comprehensive explanation of the learnable linear projection method in **Appendix F in page of 30**. Additionally, we have conducted experiments comparing learnable linear projection with random projection, demonstrating the superiority of the learnable approach in terms of convergence speed and final performance.
>
> ### **Questions:**
>
> > Q1: **Could the authors provide the training curves of the NLP experiments as shown in the CV experiments? It helps to verify if the proved fast convergence is also applicable to transformer training.**
> >
>
> **Our Response:**
>
> Thank you for this insightful suggestion. We have included the training curves for the NLP experiments in **Figure 7 at the page of  37** of the revised manuscript. These curves display both the training loss and validation perplexity across epochs for models initialized with **SAIL** compared to those with random initialization.
>
> Additionally, we have conducted a comprehensive comparative analysis of SAIL against weight transformation and linear model merging approaches, with detailed results presented in Tables 1, 2, and 3 of the General Response section. The comparative metrics demonstrate superior performance across multiple evaluation criteria. These empirical findings corroborate that the rapid convergence properties initially observed in our computer vision (CV) experiments generalize effectively to transformer-based Natural Language Processing (NLP) architectures.

---

> > ### Comment · Reviewer_1vMa · 2024-11-25
> > **Thank you for your rebuttal**
> >
> > I want to thank the authors' rebuttal, which addressed my concerns and provided more solid evidence. I have increased my score from 5 to 6.

---

> > > ### Author Response · Authors · 2024-11-26
> > >
> > > We greatly appreciate your feedback and consideration. We would like to share some new results that we have recently obtained, which are detailed in the general comments part. We hope these additional results will be of interest and provide further insights for you.

---

### Official Review · Reviewer_yrJd · 2024-11-04

**Soundness:** 3
**Presentation:** 3
**Contribution:** 2
**Rating:** 6
**Confidence:** 3

**Summary:**

This paper presents SAIL, a method to accelerate training by leveraging knowledge from pre-trained models using a structured initialization approach. It introduces a parameter transformation technique to adapt pre-trained model dimensions to the target architecture, alongside a proximal parameter integration strategy. Theoretical guarantees show the benefits of using transformed pre-trained parameters for faster convergence compared to random initialization as well as the guidance on obtaining the optimal parameters in the integration strategy. Experimental results across NLP and computer vision tasks confirm that SAIL reduces training time while improving model performance, supporting the efficacy and broad applicability of structured initialization for efficient large model training.

**Strengths:**

1. This paper offers strong motivation for addressing the challenges of efficient model initialization by harnessing pre-trained models, making a compelling case for its approach.
2. Rigorous theoretical analysis and effective visualizations are used throughout, solidly supporting the paper’s claims and enhancing interpretability.
3. Extensive experiments demonstrate that SAIL significantly outperforms random initialization, highlighting its effectiveness in reducing training time and improving model performance.

**Weaknesses:**

1. A major concern is the limited comparison with related methods. This paper’s approach aligns closely with areas like model reuse/expansion and model merging, both mentioned in the related work. Model expansion combined with proximal parameter integration or parameter transformation with model merging could potentially address the problem posed here. A comparison with methods from these areas would provide a more comprehensive evaluation of SAIL’s efficiency.
2. The paper lacks discussion on the gap between its linear model theory and real-world application, which is crucial to understanding SAIL's limitations. For instance, it would be beneficial to clarify practical computation of $\gamma^*$ and address situations where the required data isn’t available—a common issue with open-source models that often lack access to original training data.

**Questions:**

1. How do the authors justify the claim in lines 268-269 based on Theorem 2?

---

> ### Author Response · Authors · 2024-11-22
> **Official Comment by Authors (1/2)**
>
> ### Weaknesses:
>
> > W1: **Limited comparison with related methods in model reuse/expansion and model merging.**
> >
>
> **Our Response:**
>
> We appreciate your feedback regarding the need for a more thorough comparison with existing methods in model reuse, expansion, and merging. We appreciate the opportunity to elaborate on how our approach, **SAIL**, distinguishes itself from current techniques.
>
>  **The component nature of SAIL:**
>
> - **Flexibility:** Traditional model reuse and expansion methods typically initialize larger models from single smaller ones through weight replication or knowledge distillation. In contrast, **SAIL** offers a versatile framework that integrates multiple pre-trained models of diverse architectures and sizes.
> - **Parameter Transformation:**  **SAIL** employs parameter transformation techniques across both width and depth dimensions. While prior works have explored unidirectional parameter transfer strategies - either initializing larger models from smaller ones [3] or extracting subset parameters from larger models to initialize smaller ones [4] - these approaches are inherently constrained to single-direction transformations. In contrast, SAIL introduces a bidirectional parameter transformation framework that enables flexible and seamless transitions between models of varying scales.
> - **Optimal Linear Merging:** Our method introduces a systematic approach for the optimal linear merging of different models by calculating combination coefficients γ*. This process, grounded in our theoretical framework, allows for the integration of knowledge from heterogeneous models, a capability that existing methods lack.
>
> **Experiments:**
>
> - **Benchmark & Evaluation Metrics:** We have expanded our experimental evaluation to include comparisons with prominent methods such as **LIGO** [1] and **Model Soup** [2], focusing on model expansion and merging. These methods were assessed using a comprehensive suite of natural language understanding and reasoning benchmarks to ensure robust performance analysis.
> - **Results:** As presented in **General Response Tables 2 & 3**, **SAIL** demonstrates superior performance and greater flexibility compared to the evaluated existing approaches.
>
> **Ref:**
>
> [1] Wang P, Panda R, Hennigen L T, et al. Learning to grow pretrained models for efficient transformer training[J]. arXiv preprint arXiv:2303.00980, 2023.
>
> [2] Wortsman M, Ilharco G, Gadre S Y, et al. Model soups: averaging weights of multiple fine-tuned models improves accuracy without increasing inference time[C]//International conference on machine learning. PMLR, 2022: 23965-23998.
>
> [3] Du W, Luo T, Qiu Z, et al. Stacking Your Transformers: A Closer Look at Model Growth for Efficient LLM Pre-Training[J]. arXiv preprint arXiv:2405.15319, 2024.
>
> [4] Xu Z, Chen Y, Vishniakov K, et al. Initializing models with larger ones[C]//The Twelfth International Conference on Learning Representations. 2023.
>
> > W2: **The paper lacks discussion on the gap between its linear model theory and real-world application, including practical computation of γ* and situations where required data isn’t available.**
> >
>
> **Our Response:**
>
> We appreciate your feedback regarding the theoretical aspects of our work and their practical implications. In practical applications, computing the optimal weights γ* based on total variation distances between data distributions can be challenging. To address this, we have  explored alternative methods for estimating γ*:
>
> ### **Alternative Estimation Methods for $\gamma$:**
>
> To address these challenges, we propose practical estimation methods that align with our theoretical framework:
>
> ### **1. Implicit Methods Using Model Outputs or Predictions:**
>
> Even without direct access to the original datasets, we can estimate the distances between models by leveraging their outputs on a shared dataset or synthetic data. Let $\mathcal{V}$ be a validation set accessible during model deployment. We define empirical distances based on model outputs:
>
> - **Empirical Distance Between Models**:
>
>
>     $\hat{D} _{ij} = \frac{1}{|\mathcal{V}|} \sum _{x \in \mathcal{V}} \| f _{\theta_i}(x) - f _{\theta_j}(x) \|^2$
>
> - **Empirical Distance Between Models and Target**:
> $\hat{D} _{i}{D^\ast} = \frac{1}{|\mathcal{V}|} \sum _{x \in \mathcal{V}} \| f _{\theta_i}(x) - y _x^\ast \|^2,$
>
>     where $f_{\theta_i}(x)$ is the output of model $\theta_i$ for input $x$, and $y_x^\ast$ is the target output (if available).
>
>
> **Estimating $\gamma^\ast$:**
>
> Using these empirical distances, we construct an estimated matrix $\hat{H}$:
> $\hat{H} _{ij} = \hat{D} _{i}{D^\ast}^2 + \hat{D} _{j}{D^\ast}^2 - \hat{D} _{ij}^2,$
>
> and compute:
> $\hat{\gamma}^\ast = \frac{\hat{H}^{-1} \mathbf{e}}{\mathbf{e}^\top \hat{H}^{-1} \mathbf{e}}.$

---

> > ### Comment · Reviewer_yrJd · 2024-11-27
> >
> > I would like to thank the authors for their comprehensive explanations. Most of my concerns have been addressed, and I sincerely hope that these discussions will be included in the revised paper. However, I still have additional comments on the following two aspects:
> >
> > Firstly, I understand that it is challenging to directly analyze non-linear neural networks in practice. As such, providing insights from linear models is acceptable. However, it is important that the authors provide direct discussion or empirical verification (rather than indirect justification from final performance, which can be influenced by many other factors) to demonstrate how effective these insights from linear models are when applied to non-linear models.
> >
> > Secondly, regarding the claim "Theorem 2 demonstrates that initializing with the proximal parameter θ_P leads to faster convergence compared to random initialization," I believe it should be stated more cautiously. The claim should indicate that the proximal parameter is likely to achieve faster convergence, rather than stating it will definitely do so. This qualification is necessary because there is no proof that the proximal parameter strictly outperforms random initialization by reducing the difference between θ_P and θ^* by a guaranteed margin. Additionally, since the upper bound can be loose, the theoretical guarantee may not translate directly to practical performance. Therefore, the claim should be expressed with appropriate uncertainty rather than absolute certainty.

---

> > > ### Author Response · Authors · 2024-11-28
> > >
> > > ### **Our Response to First Point of Comment:**
> > >
> > > Thank you for highlighting the importance of bridging the insights from linear models to non-linear neural networks. We need to clarify that we have conducted empirical experiments to validate the key theoretical results of our SAIL in Section 4.4 and examined the training dynamics through changes in validation accuracy during training in Section 4.5.
> > >
> > > To better address this concern, we have incorporated a new section in **Appendix K at page of 43** in the revised paper. This section presents empirical validation of our theoretical findings using a non-linear model (a multi-layer perceptron with ReLU activations) on the Spirals dataset, a synthetic dataset known for its non-linear decision boundaries.
> > >
> > > In response to your comments, we examine the detailed training dynamics of our SAIL in this section, focusing on the convergence rate of the loss, among other aspects.
> > >
> > > **Conclusion of Our Empirical Verification:**
> > >
> > > **Experimental Setup.** Parameters of two MLP models, $\theta_1$ and $\theta_2$, were independently pre-trained on Spirals datasets $D_1$ and $D_2$, respectively. Utilizing SAIL, we transformed and merged these parameters to obtain the proximal parameter $\theta^{\mathrm{P}}$, which was then used to initialize a new model for training on $D^\ast$.
> > >
> > > **Baselines.** Recall that our SAIL calculates the merge ratio, denoted as $\gamma^*$, to integrate the parameters $\theta_1$ and $\theta_2$. For comparison, we manually design nine equidistant values for the merge ratio $\gamma$ within the range $[-1, 2]$. Additionally, a randomly initialized model serves as a fundamental baseline.
> > >
> > > **Key Results:**
> > >
> > > 1. **Faster Convergence:**
> > >     - The SAIL-initialized model $\theta^{\mathrm{P}}$ consistently achieves lower training loss at each epoch compared to the randomly initialized model. The training loss curve for SAIL-initialized models drops more rapidly. Additionally, our optimal $\gamma^\ast$ approximately aligns with the best-performing $\gamma$ values in loss reduction.
> > > 2. **Stable Optimization Trajectory:**
> > >     - The gradient norm for the SAIL-initialized model remains consistently lower during the initial epochs, indicating more stable and efficient updates. The gradient norms of SAIL-initialized models are significantly reduced compared to those with random initialization and those across different $\gamma$ values.
> > > 3. **Parameter Proximity:**
> > >     - The proximal parameter $\theta^{\mathrm{P}}$ constructed through SAIL is closer to the optimal parameters $\theta^\star$ than random initialization. This proximity is quantitatively supported by the inequality:
> > >     $\| \theta^{\mathrm{P}} - \theta^\star \|^2 \leq \alpha \| \theta_{\text{Random}} - \theta^\star \|^2$
> > >     where $\alpha < 1$, demonstrating that $\theta^{\mathrm{P}}$ resides closer to $\theta^\star$ in the parameter space.
> > > 4. **Optimal $\gamma$ Selection:**
> > >     - Our experiments systematically explored the impact of the merge ratio $\gamma$ on training performance. Empirically, the optimal $\gamma^\ast$ not only minimized the training loss and maximized accuracy but also maintained the lowest gradient norms. This empirical optimal $\gamma^\ast$ aligns closely with our theoretical predictions.

---

> ### Author Response · Authors · 2024-11-22
> **Official Comment by Authors (2/2)**
>
> ### **2. Model State-Based Estimation:**
>
> We can exploit internal representations to estimate model similarities:
>
> - **Activation-Based Distance**:
>
>     For each model $\theta_i$ and input $x$, let $A_{\theta_i}(x)$ denote the activation at a particular layer. We define:
>     $\hat{D} _{ij} = \frac{1}{|\mathcal{V}|} \sum _{x \in \mathcal{V}} \| A _{\theta_i}(x) - A _{\theta_j}(x) \| ^2.$
>
>
> ### **3. Supervised Fine-Tuning (SFT) or Direct Preference Optimization (DPO):**
>
> During SFT or DPO, we can incorporate the estimation of $\gamma$ into the training objective:
>
> - **Modified Loss Function**:
> $L(\theta) = L_{\text{task}}(\theta) + \lambda \sum_{i=1}^n \gamma_i \| \theta - \theta_i \|^2,$
>
>     where $L_{\text{task}}$ is the task-specific loss, and $\lambda$ is a regularization parameter.
>
>
> **Optimization Strategy:**
>
> - The coefficients $\gamma_i$ can be treated as learnable parameters, optimized jointly with $\theta$.
> - Constraints $\gamma_i \geq 0$ and $\sum_{i=1}^n \gamma_i = 1$ can be enforced using projection methods or by parameterizing $\gamma$ using a softmax function over unconstrained variables $\eta_i$:
> $\gamma_i = \frac{e^{\eta_i}}{\sum_{j=1}^n e^{\eta_j}}.$
>
> ### **Questions:**
>
> > Q1: **How do the authors justify the claim in lines 268-269 based on Theorem 2?**
> >
>
> **Our Response:**
>
> We offer a comprehensive analysis in **Appendix C** in the page of 22 by first providing an intuitive overview of how proximal parameter initialization  $\theta^P$  reduces the initial distance to the optimal parameter  $\theta^\star$ , thereby lowering the suboptimality of the loss function as indicated by **Theorem 2**. Specifically, we establish that:
> $\mathcal{J}(\theta^P) - \mathcal{J}(\theta^\star) \leq \frac{L}{2} \| \theta^P - \theta^\star \|_2^2$
>
> This demonstrates that initializing with  $\theta^P$  controls the initial loss difference through the parameter distance  $\| \theta^P - \theta^\star \|_2^2$ .
>
> Following the intuitive explanation, we present a detailed proof that begins by bounding the parameter distances using **Theorem 3**, which provides a probabilistic guarantee that pre-trained parameters are closer to  $\theta^\star$  than random initialization:
> $\Pr \left( \| \theta _i - \theta^\star \| _2^2 \leq \alpha \| \theta _{\text{rand}} - \theta^\star \| _2^2 \right) \geq 1 - O\left( \frac{\tau^2 + \beta}{\alpha} \right)$
>
> By selecting an appropriate  $\alpha$ , we ensure that $\| \theta ^P - \theta ^\star \| _2 ^2$  is sufficiently smaller than $\| \theta
> _{\text{rand}} - \theta ^\star \| _2^2$. We then relate these parameter bounds to the loss function's suboptimality using the smoothness and strong convexity properties, ultimately showing that:
> $\rho = \frac{\mathcal{J}(\theta ^P) - \mathcal{J}(\theta ^\star)}{\mathcal{J}(\theta _{\text{rand}}) - \mathcal{J}(\theta ^\star)} \leq \frac{1}{2}$
>
> This ratio confirms that  $\theta^P$  leads to a loss reduction that is at least half as effective as random initialization, thereby ensuring faster convergence of the gradient descent algorithm.

---

> ### Author Response · Authors · 2024-11-28
>
> ### **Our Response to Second Point of Comment:**
>
> Thank you for your valuable feedback. We completely agree with your point regarding the cautious phrasing of our claim. As you suggested, the statement about faster convergence with proximal parameter initialization should indeed be expressed with more uncertainty. Specifically, as stated in Theorem 2, the advantage of initializing with the proximal parameter $\theta^\text{P}$ in terms of convergence speed is likely to occur, but it holds with high probability rather than as a guaranteed result. To address this, we revised the main text (lines 267 to 273) to clearly indicate that the convergence advantage of $\theta^\text{P}$ is probabilistic, and we provided an explanation to this effect. In Appendix C, we further provide a mathematical correction to our previous proof to explicitly account for the probabilistic nature of this result.
>
> Regarding your second point, we would like to clarify that:
>
> - While we can assert that the loss function is likely advantageous initially with high probability—due to the convexity assumptions and the specific choice of $\alpha$ in Theorem 1—there are limitations in maintaining this theoretical advantage throughout the entire iteration process. We cannot guarantee a consistent lower bound on the loss function after $T$ iterations. The upper bound provided earlier is useful for initial stages but may not remain tight as iterations increase. We have clarified this in Appendix C, discussing the potential reduction of the initial advantage over time.
> - Nonetheless, our method has demonstrated superior performance in various practical scenarios. Specifically, it significantly accelerates training in neural networks across different domains, such as natural language processing (see **Section 4.4 in page of 8 and Appendix I in page of 36**), image recognition (see **Section 4.5 in page of 10 and Appendix H in page of 35**), and image generation (see **Appendix J in page of 40**).
>
> We sincerely appreciate the reviewer's insightful suggestions and valuable feedback on both the empirical verification and theoretical aspects of our article. Your constructive comments have greatly contributed to enhancing the quality and clarity of our work. Thank you again for your thoughtful and detailed review. Should you have any further questions or require additional clarification, please do not hesitate to ask.

---

> > ### Author Response · Authors · 2024-12-02
> >
> > Dear Reviewer,
> >
> > As the deadline approaches, we kindly request your response to our feedback at your earliest convenience. Thank you for your time and consideration.
> >
> > Best regards

---

> > > ### Author Response · Authors · 2024-12-03
> > >
> > > Dear Reviewer,
> > >
> > > We sincerely appreciate your thorough review of our manuscript. While we have already undertaken new empirical investigations and theoretical refinements based on your feedback, we would welcome the opportunity to engage in further discussion during the remaining time in the discussion phase. Our aim is to ensure that our revisions fully address your concerns.
> > >
> > > Best regards,
> > > Authors

---

### Author Response · Authors · 2024-11-22

# General Response

We have conducted extensive experiments to evaluate the performance of our proposed method, **SAIL**, compared to various baseline methods. Due to space constraints, we provide a summary here and include detailed settings and results in **Appendix I in page of  34**.

**Comparison Methods:**

- **Training from Scratch:** Models initialized randomly and trained on the target tasks.
- **Existing Methods:** We compared SAIL with methods such as **LIGO** (a weight transformation method) and **Model Soup** (a linear model merging method).

**Benchmarks:**

We evaluated our approach on a diverse set of NLP benchmarks to assess its performance across various linguistic and reasoning tasks:

- **PIQA:** Assesses physical commonsense reasoning.
- **HellaSwag:** Challenges models with complex multiple-choice questions requiring robust inference.
- **Winogrande:** Focuses on pronoun resolution for contextual understanding.
- **SciQ:** Tests comprehension of scientific texts.
- **ARC-Easy:** Presents grade-school level science questions for factual knowledge application.
- **COPA:** Measures causal reasoning by selecting plausible alternatives.

**Overview of Results:**

- **Table 1:** Compares the accuracy of models trained from scratch versus models initialized with SAIL across various NLP benchmarks.
- **Table 2:** Compares SAIL with LIGO, highlighting the effectiveness of our parameter transformation method.
- **Table 3:** Presents a comparison between Uniform Soup, Greedy Soup, and SAIL, demonstrating that SAIL's γ* yields better performance.

*Note: The best performance for each dataset is highlighted in bold.*

---

Table 1: Training from Scratch vs. SAIL (Accuracy)

| Dataset | Training from Scratch (%) | SAIL (Ours) (%) |
| --- | --- | --- |
| PIQA | 51.96 | **61.92** |
| HellaSwag | 24.87 | **34.48** |
| Winogrande | 51.14 | **52.96** |
| SciQ | 22.10 | **70.30** |
| ARC-Easy | 27.54 | **42.63** |
| COPA | 58.00 | **63.00** |

Table 2:  LIGO vs. SAIL (Accuracy)

| Dataset | LIGO (%) | SAIL (Ours) (%) |
| --- | --- | --- |
| PIQA | 52.29 | **61.92** |
| HellaSwag | 25.33 | **34.48** |
| Winogrande | 50.20 | **52.96** |
| SciQ | 23.90 | **70.30** |
| ARC-Easy | 29.65 | **42.63** |
| COPA | 55.00 | **63.00** |

Table 3: Uniform Soup vs. Greedy Soup vs. SAIL (Accuracy)

| Dataset | Uniform Soup (%) | Greedy Soup (%) | SAIL (Ours) (%) |
| --- | --- | --- | --- |
| PIQA | 54.80 | 57.73 | **61.92** |
| HellaSwag | 25.25 | 27.65 | **34.48** |
| Winogrande | 50.51 | 52.09 | **52.96** |
| SciQ | 51.90 | 59.70 | **70.30** |
| ARC-Easy | 27.19 | 34.91 | **42.63** |
| COPA | 57.00 | 51.00 | **63.00** |

*Note: The best performance for each dataset is highlighted in bold. For perplexity, lower is better (indicated by ↓).*

---

### Author Response · Authors · 2024-11-25

Dear Reviewers,

Based on your valuable feedback, we have expanded our experiments to enhance the training efficiency of diffusion models.

Specifically, building on the theoretical foundations of our Structured-Initialization Learning (SAIL) approach, we have discovered that a slightly modified version of SAIL can effectively **leverage representation models (e.g., DINOv2 [1]) to significantly accelerate the training of the SiT diffusion model [2]**, a state-of-the-art generative model.

For benchmarking, we conducted experiments comparing our method to the recently proposed state-of-the-art diffusion model acceleration method, REPA [3]. This method employs a pre-trained representation model DINOv2 to accelerate diffusion model training by $17.5\times$ through representation alignment, garnering significant attention in the generative model community. Detailed experimental setups are provided in Appendix J on page 39.

**Table**: Comparison of the performance of our SAIL with the baseline SiT-B/2 model and SiT-B/2 augmented with REPA across various training iterations for ImageNet $256 \times 256$ generation.

| Model | #Params | Iter. | FID↓ | IS↑ | Prec.↑ |
| --- | --- | --- | --- | --- | --- |
| SiT-B/2 | 130M | 400K | 33.0 | 43.7 | 0.53 |
| REPA  | 130M | 50K | 78.2 | 17.1 | 0.33 |
| SAIL(ours) | 130M | 50K | **67.6** | **20.5** | **0.34** |
| REPA  | 130M | 100K | 49.5 | 27.5 | 0.46 |
| SAIL(ours) | 130M | 100K | **35.9** | **45.1** | **0.53** |
| REPA | 130M | 200K | 33.2 | 43.7 | 0.54 |
| SAIL(ours) | 130M | 200K | **19.78** | **81.9** | **0.64** |
| REPA | 130M | 400K | 24.4 | 59.9 | 0.59 |
| SAIL(ours) | 130M | 400K | **12.16** | **119.4** | **0.70** |

The experimental results in the table demonstrate that our SAIL significantly outperforms both REPA and SiT.

[1] Oquab, Maxime, et al. "Dinov2: Learning robust visual features without supervision." *arXiv preprint arXiv:2304.07193* (2023).

[2] Ma, Nanye, et al. "Sit: Exploring flow and diffusion-based generative models with scalable interpolant transformers." *arXiv preprint arXiv:2401.08740* (2024).

[3] Yu, Sihyun, et al. "Representation alignment for generation: Training diffusion transformers is easier than you think." *arXiv preprint arXiv:2410.06940* (2024).

---

### Meta-Review · Area_Chair_YF28 · 2024-12-18

**Metareview:**

This submission discusses the resource challenges of developing and deploying LLMs. To address this, they introduce Sail, a method that accelerates training by leveraging publicly available pre-trained models. Sail combines a parameter transformation technique for aligning pre-trained model parameters with target architectures and a proximal parameter integration strategy for efficient model initialization, significantly reducing training time and resource usage while maintaining or improving performance on downstream tasks. However, most experiments are conducted on small-scale benchmarks. It is very hard to argue that the nanoGPT is a LLM. We see some mismatch between the claim and the experimental validation.

Most reviewers acknowledge the motivation, the importance of the targeted problem, and the theoretical analysis. However, the reviewer like the most positive reviewer seems mainly attracted by the LLM story ("This paper addresses an important problem of LLM reusing, which lacks efficient solutions before. The proposed method seems to be better than naive averaging weights.") and does not have in-depth analyses. One of the reviewers argued about the limited innovation. In our view, the major limitation of this submission lies in the insufficient large-scale experimental validation and the mismatch between claimed and validated. The models that "have revolutionized natural language processing" are ones with at least billion-level parameters.

Meanwhile, more comparisons with merging literature/methods (e.g., https://github.com/arcee-ai/mergekit) could greatly strengthen this submission, since merging serves as a key component in the proposed framework.

Lastly, in the current shape, both the abstract and introduction emphasize LLMs. However, more than 50% of experiments and findings are on tool vision datasets like CIFAR. Reorganizing and reshaping the overall story and turning down the tune of LLMs could improve this submission.

**Additional Comments On Reviewer Discussion:**

The authors did a great job during the rebuttal. The paper could be further enhanced by incorporating all the discussion and providing more experimental validation with large-scale models which will match the claim.

---

### Decision · Program_Chairs · 2025-01-22

Reject